# Atmospheric dryness reduces photosynthesis along a large range of soil water deficits

Zheng Fu [1✉], Philippe Ciais [1], I. Colin Prentice [2,3,4], Pierre Gentine [5], David Makowski [6],
Ana Bastos [7], Xiangzhong Luo [8], Julia K. Green [1], Paul C. Stoy[9], Hui Yang[1] & Tomohiro Hajima[10]

Both low soil water content (SWC) and high atmospheric dryness (vapor pressure deficit, VPD) can negatively affect terrestrial gross primary production (GPP). The sensitivity of GPP to soil versus atmospheric dryness is difficult to disentangle, however, because of their covariation. Using global eddy-covariance observations, here we show that a decrease in SWC is not universally associated with GPP reduction. GPP increases in response to decreasing SWC when SWC is high and decreases only when SWC is below a threshold. By contrast, the sensitivity of GPP to an increase of VPD is always negative across the full SWC range. We further find canopy conductance decreases with increasing VPD (irrespective of SWC), and with decreasing SWC on drier soils. Maximum photosynthetic assimilation rate has negative sensitivity to VPD, and a positive sensitivity to decreasing SWC when SWC is high. Earth System Models underestimate the negative effect of VPD and the positive effect of SWC on GPP such that they should underestimate the GPP reduction due to increasing VPD in future climates.

[1] Laboratoire des Sciences du Climat et de l'Environnement, LSCE/IPSL, CEA-CNRS-UVSQ, Université Paris-Saclay, 91191 Gif-sur-Yvette, France. [2] Georgina Mace Centre for the Living Planet, Department of Life Sciences, Imperial College London, Silwood Park Campus, Buckhurst Road, Ascot SL5 7PY, UK. [3] Department of Biological Sciences, Macquarie University, North Ryde, NSW 2109, Australia. [4] Ministry of Education Key Laboratory for Earth System Modeling, Department of Earth System Science, Tsinghua University, Beijing 100084, China. [5] Department of Earth and Environmental Engineering, Columbia University, New York, NY 10027, USA. [6] Unit Applied mathematics and computer science (UMR 518) INRAE AgroParisTech Université Paris-Saclay, Paris, France. [7] Department Biogeochemical Integration, Max Planck Institute for Biogeochemistry, D-07745 Jena, Germany. [8] Department of Geography, National University of Singapore, Singapore, Singapore. [9] Department of Biological Systems Engineering, University of Wisconsin–Madison, Madison, WI, USA. [10] Research Center for Environmental Modeling and Application, Japan Agency for Marine-Earth Science and Technology, 3173-25 Showamachi, Kanazawaku, Yokohama 236-0001, Japan. ✉email: zheng.fu@lsce.ipsl.fr

Drought poses an increasing threat to people and ecosystems around the world[1,2]. Both decreased soil water content (SWC) and increased atmospheric water demand (vapor pressure deficit, VPD) can negatively affect terrestrial gross primary production (GPP)[3–8]. Plants regulate stomatal conductance to maximize carbon gains while reducing water loss in response to high VPD[9]. Decreased SWC below a critical stress level further reinforces stomatal closure and impairs the hydraulic transfer from soils to leaves[10]. Recent studies evaluating the importance of VPD and SWC in controlling GPP or canopy conductance ($G_c$) have produced conflicting results regarding the relative roles of these two drivers[3–8], leaving it unclear how a changing water cycle will impact the carbon cycle. The key difficulty is that VPD and SWC covary due to land-atmosphere feedbacks[11,12]. Here, we use an Artificial Neural Network (ANN)[13] to separate the sensitivities of GPP, stomatal behavior and photosynthetic rates to SWC and VPD, based on daily data from flux tower observations with global coverage and a new European dataset that captures a recent extreme drought. Our aims are (1) to assess under what SWC and VPD conditions is GPP most negatively affected by droughts, (2) to test the hypothesis that GPP reduction induced by partial stomatal closure in response to decreasing SWC is partly compensated by increased photosynthetic rates to maintain carbon fixation, and (3) to evaluate whether Earth System Models (ESMs) capture the relative influence of VPD and SWC on GPP. The third aim is important because future projections of the land carbon sink depend on how models capture the response of GPP to atmospheric and soil droughts: increased exposure of plants to higher VPD from warming and drier continental relative humidity is inevitable and widespread[14], whereas changes in rainfall leading to SWC deficits vary across regions[15].

The mega-drought in the summer of 2018 over Europe was monitored by the Integrated Carbon Observation System (ICOS) network of eddy-covariance (EC) observations[16], providing an opportunity to study how GPP reacted to extremely low SWC and extremely high VPD, unobserved previously at these locations. Thus, we first analyzed time series including and excluding the year 2018 to prove the existence of nonlinear responses of GPP. Then, ANNs were trained on daily observations from EC flux towers worldwide to quantify the nonlinear sensitivities of GPP to VPD and SWC, accounting for temperature and radiation effects (Methods). To parse the observed GPP response, the same approach was then used on observation-derived canopy conductance, maximum assimilation rate and maximum carboxylation rate (Methods), and also applied to the output of ESMs participating in the Coupled Model Inter-comparison Project Phase 6 (CMIP6). We used five ESMs that reported daily output, with GPP, VPD, SWC, temperature and incoming shortwave radiation simulated by each model.

## Results and discussion

**Response of GPP to SWC and VPD.** The summer of 2018 saw the most severe summertime drought recorded in Europe during the past two decades[17]. Data from 15 EC sites with observations during 2014–2018 (Supplementary Table 1) confirmed the prevalence of exceptionally low SWC and exceptionally high VPD in 2018 (Fig. 1a, Supplementary Fig. 1). The summer average SWC was 25 (±5)% (±standard error, $n = 15$ sites) lower than in 2014–2018 and the summer average VPD was 22 (±4)% larger (Fig. 1a, Supplementary Fig. 1b), resulting in a summer GPP in 2018 that was 15 (±5)% lower than average. Low SWC conditions were often associated with high VPD (Fig. 1a). At the daily scale, there were more days with low SWC anomalies (Fig. 1b) and high VPD anomalies (Fig. 1c) during 2014–2018 than 2014–2017. We

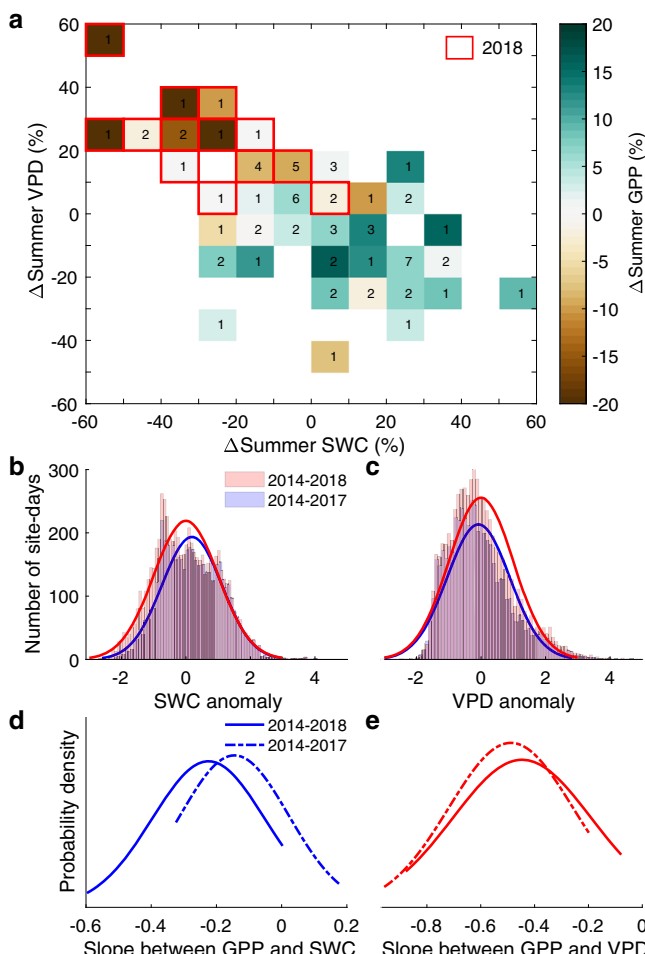

**Fig. 1 Response of gross primary production to soil water content and vapor pressure deficit. a** The response of the relative changes of summer gross primary production (GPP) to the relative changes of summer soil water content (SWC) and vapor pressure deficit (VPD) during 2014–2018. The observations from 2018 are distributed in the highlighted boxes with red borders while the 2014–2017 are mainly distributed in the other boxes (see also Supplementary Fig. 1). The number indicates the number of sites in each bin. **b–c** Histogram with a distribution fit of the number of site-days for daily SWC (**b**) and VPD (**c**) anomalies during the summer across 2014–2018 and 2014–2017. **d–e** Probability distributions across all sites for the linear regression slope of daily GPP anomalies to SWC (**d**) and VPD anomalies (**e**) during the summer across 2014–2018 and 2014–2017. The negative sign for the slope between GPP and SWC means that GPP is reduced when soil becomes drier, while the negative sign for the slope between GPP and VPD means that GPP is reduced when VPD increases.

first examined the sensitivity of daily GPP anomalies (z-scores) to SWC and VPD anomalies across 2014–2018 and 2014–2017 using multiple linear regression, accounting for SWC and VPD and their interactions at each site (Methods). Across all sites, we found a mean linear regression slope of −0.22 (−0.14 to −0.31: 95% confidence intervals) of GPP for a unit (one standard deviation) reduction of SWC during 2014–2018 (Fig. 1d). When removing the extreme drought of 2018, however, the negative mean linear regression slope with respect to SWC was reduced to −0.14 (−0.06 to −0.23). For VPD, in contrast, the linear regression slope was not significantly different whether 2018 was included (−0.45, −0.32 to −0.58) or not (−0.49, −0.37 to −0.60, Fig. 1e), emphasizing a more stable response to VPD than to SWC during extreme droughts. The positive linear regression

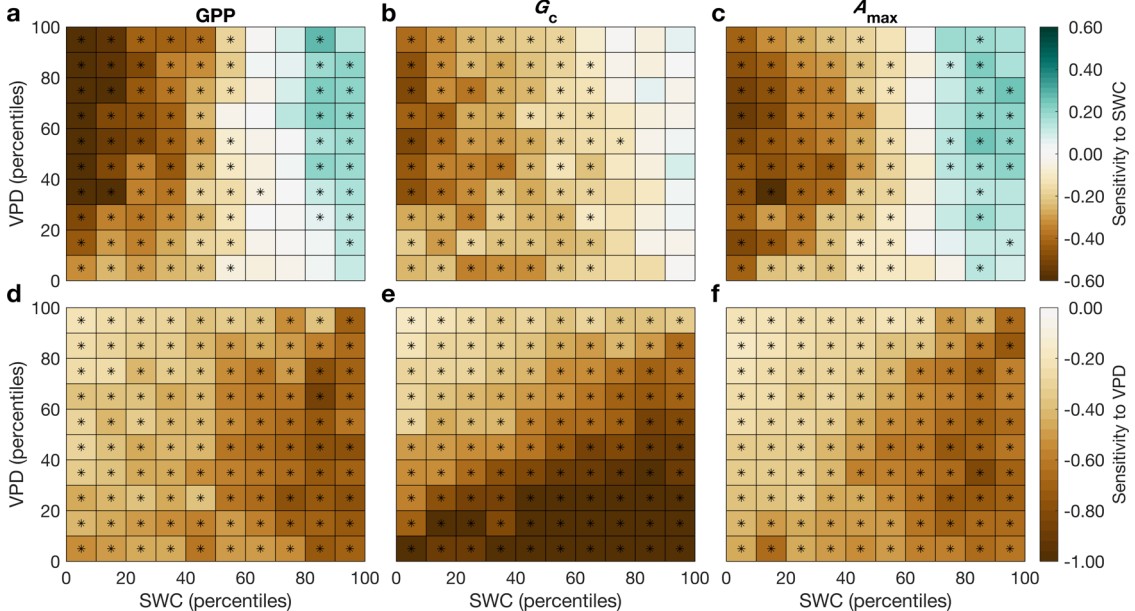

**Fig. 2 The sensitivity of gross primary production, canopy conductance and maximum photosynthetic assimilation rate to soil water content and vapor pressure deficit. a–c** The sensitivity of gross primary production (GPP), canopy conductance ($G_c$) and maximum photosynthetic assimilation rate ($A_{max}$) to soil water content (SWC). **d–f** The sensitivity of GPP, $G_c$ and $A_{max}$ to vapor pressure deficit (VPD). The percentiles are the values of 10th, 20th, …, and 90th percentile of SWC or VPD. Negative signs for the sensitivities to SWC mean GPP, $G_c$ or $A_{max}$ are reduced when SWC becomes drier while positive signs mean GPP, $G_c$ or $A_{max}$ increases when SWC becomes drier. Negative signs for the sensitivities to VPD mean GPP, $G_c$ or $A_{max}$ are reduced when VPD increases. '*' represents the sensitivities are significantly different from zero by $t$-tests ($p < 0.05$) across all sites for each bin. The number of sites in each bin were shown in the Supplementary Fig. 4.

slope between GPP and air temperature was slightly lower in 2014–2018 than in 2014–2017 while they were similar for incoming shortwave radiation (Supplementary Fig. 2).

This illustrative analysis with a linear model revealed a nonlinear sensitivity of GPP to SWC, with a disproportionate negative effect of decreasing SWC under very dry soils conditions, and a more constant sensitivity of GPP to VPD. Thus, nonlinear analysis must be used. Yet, it is not clear if different sensitivities to SWC and VPD under dry and wet soils prevail globally. To test whether this response pattern is generalizable, we combined the ICOS European EC data with the global FLUXNET2015 dataset, and calculated the sensitivities of GPP to SWC and VPD using the nonlinear ANN models[13]. To illustrate this, we gave an example and showed the ANN outputs obtained at the FR-LBr site (Supplementary Fig. 3). Results showed that predicted GPP from ANNs tracked the observed GPP well for training, testing, and validation (Supplementary Fig. 3a, b, c). At this site, the negative sensitivity of GPP to SWC deficits increases when soils get drier, whereas the negative sensitivity to high VPD prevails across the full range of SWC (Supplementary Fig. 3e, f). Next, we systematically calculated the sensitivities of GPP to SWC and VPD using ANNs at each site in the ICOS European EC data and the global FLUXNET2015 dataset; the median values were used for each bin across all sites (Methods). Consistent with evidence from the mega-drought of 2018 in Europe, we found that drought significantly increased the negative sensitivity of GPP to SWC when SWC hits its lower percentiles (<70th percentiles), whereas the sensitivity to VPD was rather insensitive to the percentile of background SWC upon which a VPD increase occurs (Fig. 2a, d). To further test the robustness of our results, we performed the same analysis separately for different plant functional types, which consistently yielded similar results (Supplementary Fig. 5). We also repeated our analysis using the SWC measurements from deep soil layers instead of the first layer (Methods). The patterns

of GPP sensitivity to SWC and VPD using deep SWC are similar with the first layer, but we also found that there were greater negative sensitivities of GPP to both SWC decreases at dry soils and VPD increases at wet soils using the SWC in the deepest layer than in other layers (Supplementary Fig. 6). This suggests that it could cause more GPP reduction if the drought happens in deeper soil layers.

Uncertainty in GPP data mainly arises from net ecosystem $CO_2$ exchange (NEE) processing and flux partitioning methods[18,19]. For the uncertainty of partitioning methods, we repeated our analysis using GPP from the daytime partitioning method[20], and compared the results obtained in our main analysis using GPP from the nighttime partitioning method[21] (Methods). The patterns of GPP sensitivity to SWC and VPD were found to be consistent between methods (Supplementary Fig. 7a–d). Across all bins, the differences in GPP sensitivity values based on the two partitioning methods mostly fell in the range from −0.1 to 0.1 (Supplementary Fig. 7e, f), indicating that flux partitioning uncertainties had minor effects on our results. Concerning NEE processing, we repeated our analysis using the quartile ranges of GPP from the nighttime partitioning method (GPP_NT_VUT_25 and GPP_NT_VUT_75, Methods), and found that the differences of GPP sensitivity values obtained between the two quartile GPPs were pretty small in most bins (Supplementary Fig. 8). Thus, our results provide robust evidence that the negative sensitivity of GPP to SWC deficits increases when soils get drier, whereas the negative sensitivity to VPD increase prevails across the full range of SWC.

**Response of canopy conductance and photosynthetic rates to SWC and VPD.** GPP changes during droughts depend on stomatal and non-stomatal (maximum photosynthetic rate) adjustments. We thus derived canopy conductance ($G_c$) and maximum photosynthetic assimilation rate ($A_{max}$) from eddy covariance

measurements (Methods) and calculated their sensitivities to SWC and VPD using ANNs. We found that the pattern of the GPP sensitivities to SWC follows that of $A_{max}$ (Fig. 2a, c), with a decrease in response to SWC dryness anomalies at low SWC conditions and an increase at high SWC (>70th percentiles). The increase of $A_{max}$ reported in relatively high SWC conditions explains why a decrease of SWC results in enhanced GPP when SWC is high. In contrast, the pattern of GPP sensitivities to VPD was similar to that of $G_c$ and $A_{max}$ (Fig. 2d–f), with greater negative sensitivity at lower VPD. $A_{max}$ can be related to the maximum carboxylation rate ($V_{cmax}$). To evaluate the response of $V_{cmax}$ to SWC and VPD, we calculated the leaf-internal $CO_2$ concentration ($c_i$) during the middle of the day at the flux towers and then derived $V_{cmax}$ (Methods). We found that the patterns of $V_{cmax}$ sensitivity to SWC and VPD are similar to those of $A_{max}$ and GPP (Supplementary Fig. 10). GPP, $A_{max}$ and $V_{cmax}$ sensitivities to SWC thus all become more negative as SWC decreases and VPD increases, but they are positive at high SWC (Fig. 2a, c, Supplementary Fig. 10).

The above analysis shows that when soils are wet, moderate soil drying is in fact accompanied by an increase in GPP. Indeed, moderate soil drying of wet soils might help increase photosynthetic biochemical activity and nitrogen uptake[22]. Experimental studies at the species level have documented that waterlogging decreased the rate of photosynthesis[23,24], the activity of Rubisco[25], and chlorophyll fluorescence[26]. Waterlogging could also decrease nitrogen availability due to leaching or denitrification, and increase exposure to toxic compounds and disease organisms[22,27,28], reducing photosynthesis. Global meta-analysis has also shown that moderate soil drying increases foliar and root nitrogen concentrations, with upregulation of root primary metabolism[29]. Here we found that, under high SWC conditions, plants also increase their carboxylation capacity in response to moderate soil drying, compensating for partial stomatal closure, allowing them to continue to assimilate $CO_2$ at high rates. Consistent with previous studies at the species level[30,31], our results indicate that there is a strong positive coupling between $G_c$ and $A_{max}$ (or $V_{cmax}$) at low SWC at the canopy scale, and at short time scales. However, we also found that this coupling is reduced for wet soils. Such behavior is consistent with an observed trade-off between $G_c$ and $V_{cmax}$ across climate gradients[30,31]. For example, Wright, Reich[31] found that species from low-rainfall environments operate (on long time scales) with substantially enhanced leaf N per unit leaf area ($N_{area}$); the higher $N_{area}$ is associated with a greater drawdown of $c_i$, such that low-rainfall species achieve higher photosynthetic rates at a given stomatal conductance. We also noted that the sensitivity of $G_c$ to decreasing SWC under wet-soil conditions is insignificant (Fig. 2c). There are two possible reasons for this. First, decreasing SWC in wet soils has negligible effect on canopy water potential so that there is no significant impact on $G_c$[32]. Second, species-specific effects may be involved. For example, Rasheed-Depardieu, Parelle[24] reported that waterlogging in *Quercus petraea* and *Quercus robur* decreased stomatal conductance while Yordanova and Popova[25] showed that there were no significant changes in stomatal conductance for maize plants.

Both GPP and $G_c$ have negative sensitivity to increasing VPD across the full range of VPD and SWC (Fig. 2d, e) while their negative sensitivity to decreasing SWC mainly occurs in a restricted range of low SWC values (Fig. 2a, b). These results are consistent with stomatal closure responses documented at leaf scale, and with plant hydraulic theory[32,33]. Stomatal closure limits decreases in water potential in the plant, ensuring that water demand from the leaves does not exceed the supply capacity of the hydraulic system—which could lead to embolism of the vascular system and even, potentially, complete desiccation of the plant. Stomatal closure responds tightly and early to leaf/canopy

water potential[33], thus increasing VPD triggers stomatal closure for the full range—as it affects water potential directly through transpiration. However, SWC only acts at relatively low values, i.e., below a threshold, given the nonlinear relationship between SWC and soil and plant water potential[32].

Our analysis considering different plant functional types consistently supports our global results (Supplementary Figs. 5, 11–12). Grasslands and savannas show a more negative sensitivity of GPP to decreasing SWC than broadleaved deciduous forests (DBF) and evergreen needle-leaved forests (ENF, Supplementary Fig. 5), which may be because forests can access to moisture in deeper soils and therefore have stronger resistance to drought[34–36]. In DBF and ENF, both GPP and $A_{max}$ sensitivities to decreasing SWC are positive when SWC is intermediate to high, but in grasslands and savannas, positive GPP and $A_{max}$ sensitivities to decreasing SWC occur only during wet-soil conditions (Supplementary Figs. 5, 12). This difference suggests that the SWC threshold for investment in high carboxylation rates may differ between forest and non-forest ecosystems. We also noted that $A_{max}$ had negative sensitivities to SWC in some high SWC bins (Supplementary Fig. 12), though most of these negative values were not statistically significant. Uncertainties in the sensitivity of GPP, $G_c$ and $A_{max}$ to water stress can be caused by species-specific trade-offs between transpiration and vulnerability to hydraulic failure[34,37].

To further consider and estimate the uncertainties of our results, we quantified the uncertainties of GPP, $G_c$, $A_{max}$ and $V_{cmax}$ sensitivities to SWC and VPD, respectively, by calculating the standard errors of sensitivities for each bin across all sites (Methods). We found that in most bins, the standard errors of GPP, $G_c$, $A_{max}$ and $V_{cmax}$ sensitivities to SWC or VPD were <0.1 (Supplementary Fig. 14). In a few cases, standard errors were higher (0.15–0.2), mainly in the bins with simultaneously high or low SWC and VPD (80–100th percentiles or 0–20th percentiles, Supplementary Fig. 14), where there were fewer data points (Supplementary Fig. 4).

Across all sites, the sensitivity of $G_c$ to SWC becomes more negative as SWC decreases (Fig. 2b) and is also negative under low VPD and low SWC. Combining our diagnostics of the sensitivities of $G_c$ and GPP to VPD and SWC, we calculated how ecosystem intrinsic water use efficiency (iWUE, defined by the ratio of GPP to $G_c$) changes with SWC and VPD (Supplementary Fig. 16). The positive sensitivity of iWUE to decreasing SWC is general, indicating that iWUE is enhanced when soil becomes dry. The sensitivity of iWUE to SWC is more positive under high VPD values while its sensitivity to decreasing VPD is more negative under low VPD values. These results shed light on conflicting observations at the site-scale[38,39] and confirms findings from atmospheric carbon isotopes at a large continental scale, showing that drought tends to increases iWUE[40].

**Relative roles of SWC and VPD**. Regarding the relative roles of SWC and VPD, we demonstrated that VPD dominates dryness stress on ecosystem production while SWC becomes important under dry soils. The mean linear regression slope (the standardized partial regression coefficient as all predictors were standardized) across all sites in Europe showed that VPD (−0.45 and −0.49, across 2014–2018 and 2014–2017, respectively) had larger negative effects on GPP than SWC (−0.22 and −0.14, Fig. 1d, e). Consistent with the linear model analysis in Europe, ANNs analysis found that VPD always dominates dryness stress on GPP as long as the SWC is not low, while the negative effects of decreasing SWC on GPP are larger than that of VPD under the low SWC conditions (<30th percentiles, Fig. 3a). Among different VPD gradients, the VPD effects are also always more negative

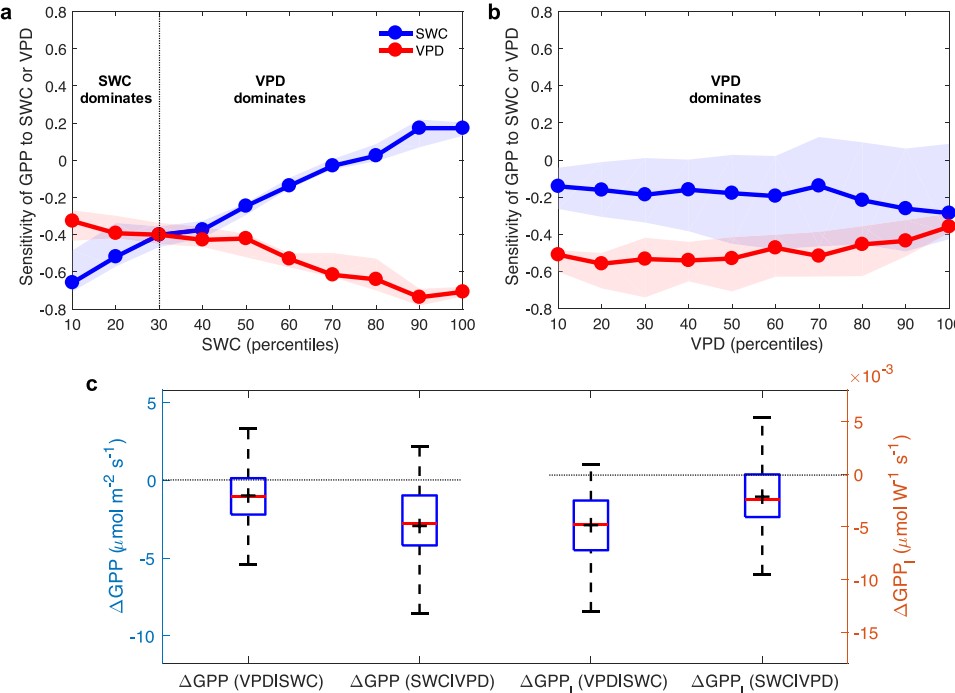

**Fig. 3 Disentangling soil water content and vapor pressure deficit limitation effects. a** The sensitivity of gross primary production (GPP) to soil water content (SWC) or vapor pressure deficit (VPD) at each SWC bin. **b** The sensitivity of GPP to SWC or VPD at each VPD bin. The solid lines indicate the median values and the uncertainty bounds refer to the range from 25th to 75th percentiles. **c** Effect of low SWC and high VPD on GPP using the approaches of Liu, Gudmundsson[5] across all sites. The terms are defined following Liu, Gudmundsson[5]. ΔGPP (VPD|SWC): VPD limitation on GPP without SWC-VPD coupling; ΔGPP(SWC|VPD): SWC limitation on GPP without SWC-VPD coupling. GPP$_I$ means GPP is normalized by incoming (I) shortwave radiation to remove the radiation effects. For each box plot, the '+' indicates the mean; the box indicates the upper and lower quartiles and the whiskers indicate the 5th and 95th percentiles of the data.

than that of SWC, although the negative effects of SWC tend to increase along with VPD increases (Fig. 3b). These results reconcile previous conflicting assessments on the roles of VPD versus SWC[3–8], because the relative importance of VPD and SWC depends on soil water conditions. In the future, warmer temperatures and lower relative humidity will further increase the relative importance of VPD in limiting ecosystem production globally[7,41].

Our findings differ from a recent global assessment of a predominant SWC stress on GPP using global solar-induced chlorophyll fluorescence (SIF) satellite observations and re-analysis climate data[5]. To investigate the possible reasons for this discrepancy, we reproduce the approach from ref. [5] at each site, and find that, while our observations indicate that low SWC reduces GPP, high VPD is more important than low SWC across all sites when we removed the radiation effects (Fig. 3c). Once neglecting the radiation effects, the negative effects of low SWC are larger than that of high VPD (Fig. 3c). We therefore suggest that the role of VPD in previous studies that neglected the strong VPD-radiation coupling should be re-visited. It is also worth noting that SIF satellite sensor passes over each land pixel just once per day (typically in the morning when VPD is relatively low)[5,42]. SIF is also less sensitive to stomatal regulation than GPP[43], which could help to explain why ref. [5] did not find significant VPD effects. The above analysis with three lines of evidence from linear, nonlinear (ANNs) model and the approach of ref. [5] consistently showes the dominant role of VPD in leading to drought limitation on vegetation productivity.

**Comparison with ESM simulations.** Last, we diagnosed the sensitivities of daily GPP to SWC and VPD from five CMIP6

ESMs (ACCESS-ESM1-5, CMCC-CM2-SR5, IPSL-CM6A-LR, NorESM2-LM and NorESM2-MM, Supplementary Table. 3), all of which provided daily outputs. We found that each of these ESMs were aligned with our observational finding that the negative sensitivity of GPP to VPD is general (Fig. 4f–j and Fig. 2). In addition, three models that used the Community Land Model captured the negative sensitivity of GPP in response to SWC dryness anomalies at low SWC conditions and a positive sensitivity at high SWC (CMCC-CM2-SR5, NorESM2-LM and NorESM2-MM, Fig. 4a–e and Fig. 2). However, all ESMs underestimated both the negative sensitivity of GPP to increasing VPD (0.19 ± 0.12, median across five ESMs ± standard error) and its positive sensitivity to decreasing SWC (−0.20 ± 0.07) at high SWC (>80th percentiles) (Fig. 4k–t). In other words, the models showed a compensation in their GPP sensitivities to VPD and SWC and provided a reasonable overall response of GPP to droughts, but not necessarily for the right reasons. Continued warming is likely to lead to different trajectories of VPD and SWC (ref. [7]), potentially leading to incorrect projections of terrestrial changes in GPP as VPD and SWC changes diverge. These results emphasize that model evaluation should carefully address covarying factors, especially during extremes. The ESMs could not account for the response found in this study by which plants increase their carboxylation capacity with moderate soil drying at high SWC, compensating for partial stomatal closure, and thus allowing $CO_2$ assimilation to continue at a high rate. This deficiency in part explains why all the ESMs underestimated the positive sensitivity of GPP to SWC decrease under wet-soil conditions. Our results also suggest that the implementation of plant hydraulics in ESMs should allow both effects (VPD and SWC) to be better represented, because plant hydraulics

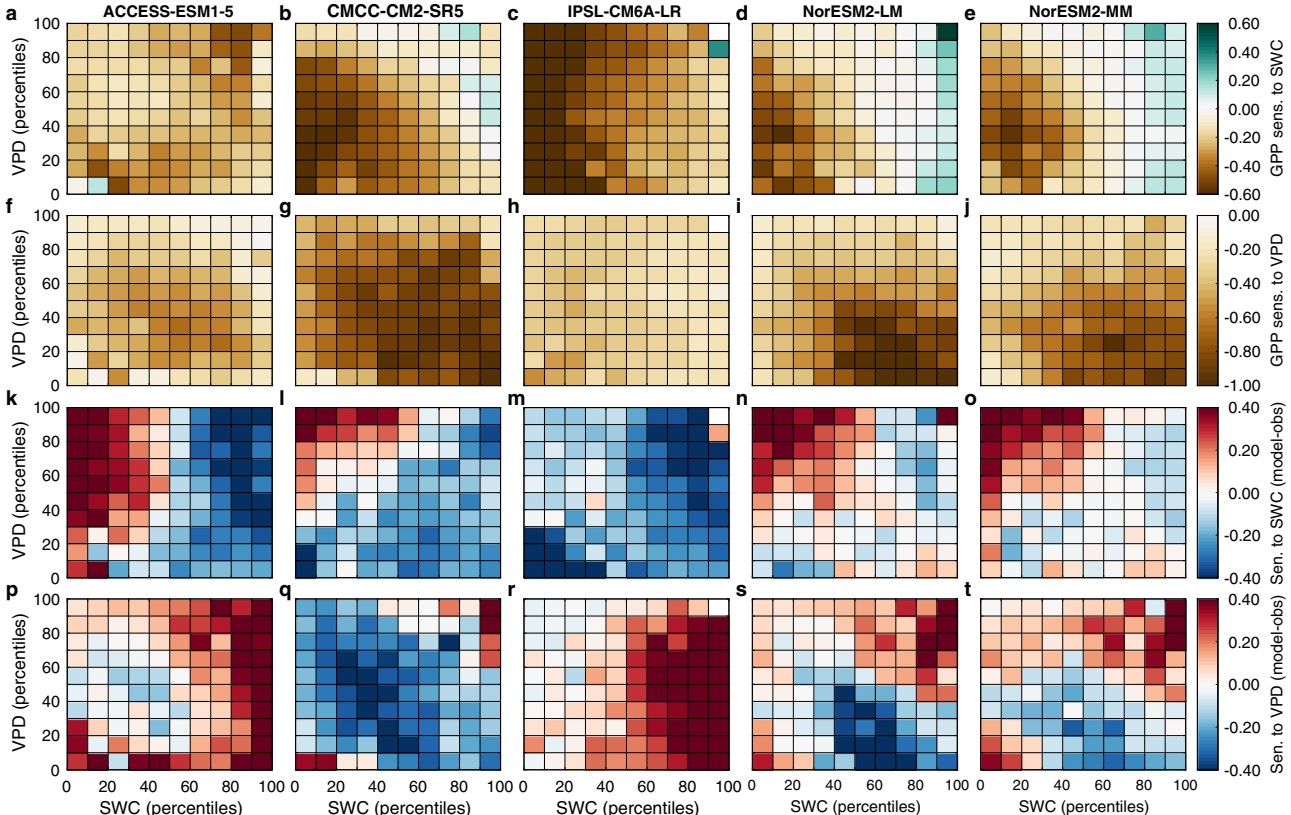

**Fig. 4 The sensitivity of gross primary production to soil water content and vapor pressure deficit using five Earth System Models. a–j** The sensitivity of gross primary production (GPP) to soil water content (SWC) and vapor pressure deficit (VPD) using five Earth System Models. **k–t** The differences (modeled sensitivity minus observation-based sensitivity) for GPP to SWC (**k–o**) and VPD (**p–t**).

play a critical role in leaf gas exchange by regulating stomatal conductance.

Understanding and quantifying the distinct responses of photosynthesis to soil and atmospheric dryness stress is essential to reliably project terrestrial ecosystems carbon uptake in a changing climate. In this study, we confirmed that low SWC and high VPD strongly decrease GPP, and provided regional and global evidence that the negative sensitivity of GPP to SWC increases as soils get drier, while its negative sensitivity to high VPD prevails across the full range of soil moisture, using both linear and nonlinear models. Thus, even when soil moisture is above the stress threshold, exposure to high VPD still causes a reduction of GPP through stomatal closure. The implication is that atmospheric drought that does not trigger SWC feedbacks can still reduce GPP in the future, as VPD increases over vegetated areas[7,41]. The pattern of GPP sensitivities to SWC follows that of $A_{max}$ while the pattern of the GPP sensitivities to VPD follows that of $G_c$ and $A_{max}$. The GPP, $A_{max}$ and $V_{cmax}$ sensitivities to SWC become more negative as SWC decreases and VPD increases but they are positive at high SWC values, suggesting that ecosystems compensate their stomatal reduction by higher carboxylation rates to continue to assimilate carbon under moderate drought, consistent with optimality theory and trade-offs between $G_c$ and $V_{cmax}$ along climate gradients. Under soil moisture deficits, carboxylation rates increase and offset the reduction of conductance, implying an increase of water use efficiency. The same analysis broken down by different plant functional types further supported our global results. Three lines of evidence highlighted the different role of VPD and SWC on ecosystem production and thus reconciled previous conflicting

assessments. The five state-of-the-art ESMs that we tested failed to accurately reproduce the magnitudes of sensitivities, underestimating them by about 0.2 for both the negative sensitivity to VPD and the positive sensitivity to SWC at high SWC levels. This indicates that current models will not accurately project the response of carbon uptake and transpiration to future droughts. Together, these results suggest that atmospheric dryness reduces photosynthesis along a large range of soil water deficits, and they highlight the importance of correctly evaluating the ecosystem-scale response to the under-appreciated exposure to atmospheric dryness as both soil and atmospheric dryness will increase with climate change.

## Methods

**Eddy-covariance observations**. We used half-hourly or hourly GPP, air temperature, VPD, SWC and incoming shortwave radiation from the recently released ICOS (Integrated Carbon Observation System)[44] and the FLUXNET2015 dataset of energy, water, and carbon fluxes and meteorological data, both of which have undergone a standardized set of quality control and gap filling[19]. Data were already processed following a consistent and uniform processing pipeline[19]. This data processing pipeline mainly included: (1) thorough data quality control checks; (2) calculation of a range of friction velocity thresholds; (3) gap-filling of meteorological and flux measurements; (4) partitioning of $CO_2$ fluxes into respiration and photosynthesis components; and (5) calculation of a correction factor for energy fluxes[19]. All the corrections listed were already applied to the available product[19]. We used incoming shortwave radiation, temperature, VPD, and SWC that were gap-filled using the marginal distribution method[21]. The GPP estimates from the night-time partitioning method were used for the analysis (GPP_NT_VUT_REF). SWC was measured as volumetric SWC (percentage) at different depths, varying across sites. We mainly used the surface SWC observations but deeper SWC measurements were also used when available. Data were quality controlled so that only measured and good-quality gap filled data (QC = 0 or 1) were used.

**Analysis of the extreme summer drought in 2018 in Europe to prove non-linearity**. To analyze the effect of summer drought in 2018 on GPP in Europe, we selected 15 sites with measurements during 2014–2018 from the ICOS dataset, representing the major ecosystems across Europe (Supplementary Table 1). Croplands were excluded due to the effect of management on the seasonal timing of ecosystem fluxes, both from crop rotation that change from year to year and from the variable timing of planting and harvesting. In croplands, the changes of GPP anomalies across different growing season could be mainly depend on crop varieties and management activities. Information of crop varieties, growing times yearly and other management data for each cropland site should be collected in future in order to fully consider and disentangle the impacts of SWC and VPD on its photosynthesis. Wetland sites were also removed because they are influenced by upstream organic matter and nutrient input, as well as fluctuating water tables. Daytime half-hourly data (7 am to 19 pm) were aggregated to daily values. At each site, the relative changes (△X) of summer (June–July–August) GPP, SWC and VPD during 2014–2018 refer to the summer average of 2014–2018 were calculated for each year. For example, the calculation of the relative change in 2018 is shown in Eq. (1):

$$\triangle X = \frac{X_{2018} - X_{average\ of\ 2014-2018}}{X_{average\ of\ 2014-2018}} \times 100\% \qquad (1)$$

where $X_{2018}$ is the mean of the daily values of X (GPP, SWC, or VPD) during the summer of 2018, and $X_{average\ of\ 2014-2018}$ is the mean of the daily values of X over all the summers of the 2014–2018 period. The average △X across a certain number of sites at each bin were used for the results in Fig. 1a.

Daily time series of GPP, SWC and VPD during summer for each site were normalized (z-scores) to derive the standardized sensitivity of GPP to SWC and VPD. For each variable, the mean value across the summer of 2014–2018 was subtracted for each day at each site and then normalized by its standard deviation. At each site, we used a multiple linear regression (Eq. 2) to estimate daily GPP anomalies sensitivities to SWC and VPD anomalies across 2014–2018 and 2014–2017, respectively:

$$GPP = \beta_1\ SWC + \beta_2\ VPD + \beta_3\ SWC \times VPD + \beta_4\ T_a + \beta_5 RAD + b + \varepsilon \qquad (2)$$

where $\beta_i$ is the standardized sensitivity of GPP to each variable; $T_a$ represents the air temperature; $RAD$ represents the incoming shortwave radiation; $b$ represents the intercept; and $\varepsilon$ is the random error term. We compared estimated sensitivities with and without 2018 data to quantify the impacts of extreme drought in 2018 on GPP sensitivity to SWC (Fig. 1d) and VPD (Fig. 1e). The slope was calculated at each site and then the distribution of slopes across sites were plotted in Fig. 1d, e.

**Global analysis of the sensitivities of GPP to SWC and VPD**. For the global analysis, instead of summer, we focused on the growing season and days when the SWC and VPD effects were most likely to control ecosystem fluxes and screen out days when other meteorological drivers were likely to have a larger influence on fluxes. Following previous studies[5,8,45], for each site, we restrict our analyses to the days in which: (i) the daily average temperature >15 °C; (ii) sufficient evaporative demand existed to drive water fluxes, constrained as daily average VPD > 0.5 kPa; (iii) high solar radiation, constrained as daily average incoming shortwave radiation >250 Wm$^{-2}$. By combining ICOS and FLUXNET2015 data, at the global scale, we evaluated 67 sites with at least 300 days observations over the growing seasons for the years available (Supplementary Table 2). We excluded cropland and wetland sites for the above-mentioned reasons. These 67 sites were used to calculate the relative effects of low SWC and high VPD on GPP following the approach of ref. [5] (see below sections). For 8 sites, the ANN results failed performance criteria (the correlation between predicted GPP and observed GPP is <0.5). The remaining 59 sites were used for ANNs and sensitivity analysis (Supplementary Table 2). At each site, each variable was first normalized to z-scores over the growing seasons for the years available, then we binned daily SWC and VPD values into 10 × 10 percentile bins and assessed the sensitivities for each bin using ANNs for each site. The median values of sensitivities across all sites were used for the results.

**Derivation of $G_c$, $A_{max}$ and $V_{cmax}$ from eddy covariance measurements**. $G_c$ during the growing season was calculated using half-hourly data (removing rainy days) by inverting the Penman–Monteith equation[46] (Eq. 3):

$$G_c = r_a\gamma / \left( \frac{\triangle(R_n - G) + \rho c_p r_a (e_s(T_a) - e_a)}{\lambda E} - (\triangle + \lambda) \right) \qquad (3)$$

where $G_c$ and $r_a$ are canopy stomatal conductance and aerodynamic resistance respectively, $\gamma$ is the psychrometric constant, $\triangle$ is the slope of the water vapor deficit with respect to temperature, $R_n$ and $G$ are observed net radiation and soil heat flux, $\rho$ is air density, $C_p$ is the specific heat capacity of dry air, $e_s$ and $e_a$ are saturated and actual vapor pressure, and $\lambda E$ is observed evapotranspiration. $r_a$ is calculated following Novick, Ficklin[7] (Eq. 4), using the von Kármán constant ($k = 0.4$), available wind speed data ($w_s$), measurement height ($z_m$), momentum roughness length ($z_0 = 0.1\ h$) and zero plane displacement

($z_d = 0.67\ h$), both based on calculated canopy height ($h$) under near-neutral conditions[47] (Eq. 5).

$$r_a = \frac{\ln\left(\frac{z_m - z_d}{z_0}\right)^2}{w_s k^2} \qquad (4)$$

$$h = \frac{z_m}{0.6 + 0.1 \times \exp\left(\frac{kw_s}{u^*}\right)} \qquad (5)$$

In order to evaluate changes in biochemical processes, we derived daily $A_{max}$ from non-gap-filled $F_c$ measurements using eddy covariance observations[48]. The instantaneous rate of photosynthesis generally increases with incoming radiation and saturates (at $A_{max}$) as illumination increases. The relationship between the instantaneous rate of photosynthesis and incoming shortwave radiation has been well documented using light response curves (LRCs)[48,49]. In the process of partitioning $F_c$ into an ecosystem photosynthesis and respiration term using the daytime partitioning method[20], a key step is to fit $F_c$ with an LRC:

$$F_c = \frac{\alpha\beta R_g}{\alpha R_g + \beta} + \gamma \qquad (6)$$

where $\alpha$ is the canopy-scale quantum yield; $\beta$ is the maximum rate of $CO_2$ uptake of the canopy at saturating light, equivalent to $A_{max}$; $R_g$ is the global radiation; and $\gamma$ is ecosystem respiration. The impact of VPD on $\beta$ is considered by requiring that $\beta$ decreases exponentially with the increase of VPD when VPD exceeds a threshold ($VPD_0$):

$$\beta = \begin{cases} \beta_0\exp(-k(VPD - VPD_0)), VPD > VPD_0 \\ \beta_0, \quad VPD \le VPD_0 \end{cases} \qquad (7)$$

where $\beta_0$ and $k$ are fitted parameters and $VPD_0$ is 1 kPa[48]. Following Luo and Keenan[48], we applied this method to a short time window (2–14 days) of $F_c$ depending on the availability of flux measurements and assumed that every day in the same time window has the same daily $A_{max}$. We retrieved the daily $A_{max}$ by implementing Eqs. (6) and (7) using the REddyProc R package (https://github.com/bgctw/REddyProc)[20].

$V_{cmax}$ represents the activity of the primary carboxylating enzyme ribulose 1,5-bisphosphate carboxylase–oxygenase (Rubisco) as measured under light-saturated conditions. To evaluate the responses of $V_{cmax}$ to SWC and VPD, we first calculated the daily internal leaf $CO_2$ partial pressure ($c_i$) in the middle of the day (11:00–14:00) via Fick's Law (Eq. 8), excluding periods with low incoming shortwave radiation (<500 W m$^{-2}$).

$$c_i = c_a - GPP \times (r_{co2} + r_a) \qquad (8)$$

where $c_a$ is the atmospheric $CO_2$ partial pressure, and $r_{co2}$ is the ecosystem resistance to $CO_2$ ($1.6/G_c$). Then we derived $V_{cmax}$ according to the standard biochemical model (Eq. 9):

$$A_{max} = V_{cmax} \frac{(C_i - \Gamma^*)}{(C_i + K)} \qquad (9)$$

where $\Gamma^*$ is the $CO_2$ compensation point in the absence of mitochondrial respiration and $K$ is the effective Michaelis–Menten coefficient of Rubisco. Both $\Gamma^*$ and $K$ are temperature-dependent variables[50]. Values of $V_{cmax}$ were standardized to 25 °C using the Arrhenius equation with activation energies from Bernacchi et al.[51,52].

**Artificial neural networks and sensitivity analysis**. ANN has been used with eddy covariance datasets[53–55] and remote sensing datasets[13,42,56] in the Earth sciences as predictive or analysis tool. We used the ANN to analyze the sensitivities of GPP, $A_{max}$, $V_{cmax}$, $G_c$ and iWUE to SWC and VPD. ANN was chosen for this application because it has nonlinear activation functions, which can effectively predict nonlinear effects[13,54,57]. We limited the ANN fit to the small number of predictors that are known environmental drivers, in order to avoid over-fitting[54]. Daily temperature, VPD, SWC and incoming shortwave radiation were used as predictor variables while daily GPP (or $G_c$, $A_{max}$, $V_{cmax}$, iWUE) is used as a response variable. Feed-forward ANN (one hidden layer) was trained using the Matlab 'Neural fitting toolbox' and repeated five times. The number of nodes in the hidden layer was sampled from 4 to 20 (step size 2), and 10 was selected because the results from different nodes were very similar. 60% of the data were used for the purpose of training the ANN while the remaining 40% of the data were used for validation (20%) and testing (20%). Performance was assessed by correlations ($r$) and root-mean-square errors. Results showed the $r$ values were >0.7 at most sites. During the training process, weight and bias values were optimized using the Levenberg–Marquardt optimization[58,59]. The maximum number of epochs to train is 1000. An example to demonstrate the ANN training at one site was shown in Supplementary Fig. 3.

At each site, ANN was run and sensitivities were calculated for all data within each SWC and VPD bin and the median value was used. For each of the five trained ANNs, one of the predictor variables was perturbed by one standard deviation (a value of 1 due to the initial input data normalization), and GPP was predicted again using the existing ANN with the predictors including the perturbed

variable; this process was repeated for each predictor variable. The predicted values of GPP obtained with and without perturbation were then compared to determine the sensitivity values. The sample equation showing the calculation of the GPP sensitivity to VPD is shown in Eq. (10).

$$\text{Sensitivity}_{VPD} = median\left(\frac{GPP_{(ANN\ VPD+stdev(VPD))} - GPP_{(ANN\ all\ VAR)}}{stdev(VPD)}\right) \quad (10)$$

We repeated the ANN and sensitivity analyses five times and the median of these were used at each site. Across all sites, significances of the sensitivities for each bin were tested using $t$-tests ($p < 0.05$). The number of sites at each bin were shown in the Supplementary Fig. 4. We defined the sensitivity sign following the change of GPP: negative sensitivity means GPP decrease while positive sensitivity means GPP increase. That is to say, negative signs for the sensitivities to SWC mean GPP, $G_c$ or $A_{max}$ are reduced when SWC becomes drier while positive signs mean GPP, $G_c$ or $A_{max}$ increases when SWC becomes drier; negative signs for the sensitivities to VPD mean GPP, $G_c$ or $A_{max}$ are reduced when VPD increases.

The uncertainty of GPP used in this study mainly arises from net ecosystem $CO_2$ exchange (NEE) processing and flux partitioning methods[18]. Concerning partitioning methods, we repeated the sensitivity analysis using GPP from the daytime partitioning method (GPP_DT)[20], and compared the results obtained in our main analysis using GPP from the nighttime partitioning method (GPP_NT)[21]. It should be noted that VPD is used as limiting factor for estimating GPP_DT, so it was a good choice to use the GPP_NT. The uncertainty from these two different partitioning methods were quantified by calculating the differences of their sensitivities (e.g., GPP_NT sensitivity to SWC minus GPP_DT sensitivity to SWC) for each bin (Eq. 11, Supplementary Fig. 7e, f). We also calculated the relative uncertainty using the absolute value of differences of sensitivities divided by the absolute value of mean sensitivities, indicating that the uncertainty represents how many percent of the mean sensitivity (Eq. 12, Supplementary Fig. 7g, h). Please note that the high levels of relative uncertainty occurred in the bins with statistically insignificant sensitivity values (Supplementary Fig. 7a, c, g). Since these sensitivity values are close to zero, a low absolute uncertainty leads to a high relative uncertainty. For the uncertainty of NEE processing, we repeated our analysis using the quartile ranges of GPP from the nighttime partitioning method (GPP_NT_VUT_25 and GPP_NT_VUT_75), which were available for all the sites in both the collections used and derived by the uncertainty in NEE. Similarly, the absolute and relative uncertainties from these two quartile GPPs were also quantified (Supplementary Fig. 8).

$$\text{Uncertainty}_{SWC} = \text{Sensitivity}_{GPP\_NT\ to\ SWC} - \text{Sensitivity}_{GPP\_DT\ to\ SWC} \quad (11)$$

$$\text{Relative uncertainty}_{SWC} = 100\% \times \left|\frac{\text{Uncertainty}_{SWC}}{(\text{Sensitivity}_{GPP\_NT\ to\ SWC} + \text{Sensitivity}_{GPP\_DT\ to\ SWC})/2}\right| \quad (12)$$

To evaluate the effects of SWC in different depths on the sensitivity of GPP to SWC and VPD, we repeated the sensitivity analysis using 31, 24, and 17 sites with SWC measurements in the second (SWC_2), third (SWC_3), and fourth (SWC_4) depths, respectively (2-4: increases with the depth, 4 is deepest). To test if the phenological cycle affects our results, we repeated our analysis (1) using only peak growing season, the 3-month period with the maximum mean GPP across the available years, where seasonal variability is muted; (2) using anomalies by removing the seasonal cycle, which was calculated by averaging all available years of the data and smoothing the series with a 30-day moving average as Feldman, Short Gianotti[60]. Both analyses yield similar results (Supplementary Fig. 9).

The main sources of uncertainty for $G_c$ is the latent heat flux uncertainty from eddy covariance measurements. We used both the 'LE' and 'LE.CORR' variables reported by the ICOS and FLUXNET2015 database for latent energy exchange. LE.CORR is the "energy balance corrected" version of latent heat flux, based on the assumption that Bowen ratio is correct. Our results were robust to either variable (Supplementary Fig. 13). The differences in $G_c$ sensitivity values based on the two latent heat fluxes mostly fell in the range from −0.1 to 0.1 (Supplementary Fig. 13e, f). The uncertainty of $A_{max}$ was evaluated by Luo and Keenan[48], who showed that the values of $A_{max}$ and $A_{2000}$ (ecosystem photosynthesis at a photosynthetic photon flux density of 2000 $\mu$mol m$^{-2}$ s$^{-1}$) were very consistent, indicating that the uncertainty in $A_{max}$ from this method is small. The effects of measurement uncertainties on $V_{cmax}$ are difficult to assess because of a lack of repetition of the measurements of the variables used to derive $V_{cmax}$. However, this source of uncertainty should not hamper our results because the measurements were done at high frequency and were automatic for all flux towers, thus with random errors mainly, and limiting the risk of bias.

In addition, to further consider and estimate the uncertainties of our results, we quantified the uncertainties of GPP, $G_c$, $A_{max}$ and $V_{cmax}$ sensitivities to SWC and VPD, respectively, by calculating the standard errors of sensitivities for each bin across all sites (Supplementary Fig. 14). We also calculated the relative uncertainties of GPP, $G_c$, $A_{max}$ and $V_{cmax}$ sensitivities to SWC and VPD, respectively, using the standard errors divided by the absolute value of median sensitivities (Supplementary Fig. 15).

**Approach of ref. [5] to disentangle the relative role of SWC and VPD on GPP**. In a recent paper, Liu et al.[5] performed a global analysis using SIF and re-analysis

climate data. They estimated the difference between SIF at the highest VPD bin and lowest VPD bin in each SWC bin to derive the ΔSIF(VPD|SWC). Similarly, SWC limitation on SIF without SWC-VPD coupling, termed ΔSIF (SWC|VPD), was derived from the changes in SIF from high SWC to low SWC at each VPD bin. Applying this approach to daily GPP and GPP_I (GPP normalized by incoming shortwave radiation (I) to limit the impact of radiation) respectively, we derived the ΔGPP (VPD|SWC), ΔGPP(SWC|VPD), ΔGPP_I (VPD|SWC) and ΔGPP_I (SWC| VPD) at each site (Supplementary Table 2).

**CMIP6 ESM simulations**. Five ESMs (ACCESS-ESM1-5[61], CMCC-CM2-SR5[62], IPSL-CM6A-LR[63], NorESM2-LM[64] and NorESM2-MM[64]) in CMIP6 provide daily GPP; most models provide only monthly GPP outputs (Supplementary Table 3). Daily GPP, air temperature, incoming shortwave radiation, surface soil moisture, and calculated VPD (from temperature and relative humidity) estimations from historical runs (1995–2014) were extracted from each model according to the site locations. Following the observational analysis, the same analysis was carried out for the five CMIP6 models. Each variable was first normalized using $z$-scores for each site over the growing season, and an ANN was created at each site for each model. Similar to the observational analysis, ANN and sensitivity analyses were performed five times and the median of these were used. At each site, sensitivities were calculated for all data within each SWC and VPD bin and each bin was summarized by its median value. The median values of sensitivities across all sites for each bin were used for the results. To evaluate the sensitivity performance in ESMs, we calculated the difference between modeled and observed sensitivities (Fig. 4).

## Data availability
The data used in this study are openly available in the following databases: The eddy covariance measurements are obtained from the ICOS (https://www.icos-cp.eu/data-products/YVR0-4898) and FLUXNET2015 datasets (https://fluxnet.fluxdata.org/data/fluxnet2015-dataset/). The CMIP6 data were downloaded from https://esgf-data.dkrz.de/search/cmip6-dkrz/.

## Code availability
The code used to calculate the $G_c$, $A_{max}$ and $V_{cmax}$ is publicly available at https://github.com/fueco/GcAmaxVcmax.

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

## Acknowledgements

This work was financially supported by the European Research Council Synergy project SyG-2013-610028 IMBALANCE-P (to Z.F. and P.C.) and the ANR CLAND Convergence Institute (to Z.F., P.C. and D.M.). I.C.P acknowledges support from the European Research Council under the European Union's Horizon 2020 research and innovation programme (grant agreement No: 787203 REALM). This work is a contribution to the Imperial College initiative on Grand Challenges in Ecosystems and the Environment (to I.C.P). P.C.S acknowledges support from the U.S. National Science Foundation award DEB-1552976. P.G. acknowledges funding from LEMONTREE Schmid futures grant and NASA grants NNH17ZDA00IN-THP and NNX16AO16H. We would like to thank the ICOS Infrastructure for support in collecting and curating the eddy covariance data for 2018. This work used global eddy covariance data acquired and shared by the FLUXNET community, including these networks: AmeriFlux, AfriFlux, AsiaFlux, CarboAfrica, CarboEuropeIP, CarboItaly, CarboMont, ChinaFlux, Fluxnet-Canada, GreenGrass, ICOS, KoFlux, LBA, NECC, OzFlux-TERN, TCOS-Siberia, and USCCC. The ERA-Interim re-analysis data are provided by ECMWF and processed by LSCE. The FLUXNET eddy covariance data processing and harmonization were carried out by the European Fluxes Database Cluster, AmeriFlux Management Project and Fluxdata project of FLUXNET, with the support of CDIAC and ICOS Ecosystem Thematic Center and the OzFlux, ChinaFlux and AsiaFlux offices.

## Author contributions

Z.F. and P.C. designed the study. Z.F. performed the analysis. Z.F. and P.C. wrote the paper with the inputs from all co-authors. I.C.P., P.G., D.M., A.B., X.Z.L., J.K.G.,

P.C.S., H.Y., and T.H. provided method suggestions and contributed to the interpretation of the results.

## Competing interests

The authors declare no competing interests.
