## [Peer Review File · Nature Communications]

Title: Atmospheric dryness reduces photosynthesis along a large range of soil water deficitsREVIEWER COMMENTS

Reviewer #1 (Remarks to the Author):

The paper assessed gross primary production (GPP) sensitivities to low soil water content (SWC) and water vapor pressure deficit (VPD) to evaluate under what SWC and VPD conditions is GPP most negatively affected by droughts. Authors found that VPD dominated dryness stress on ecosystem production while SWC becomes important only under very dry soils. Their findings in these sense, after using three different procedures (linear, nonlinear model and the approach of ref.3), differ from a recent published paper (ref.3; also in Nat. Commun., 2020) that showed a global assessment of a predominant SWC stress on GPP using solar-induced chlorophyll fluorescence satellite observations and re-analysis data. After investigating the discrepancy, authors suggested to reexamine the role of VPD in studies that neglected the strong VPD-radiation coupling.

Another important finding is that 5 CMIP6 Earth System Models (ESM) underestimate the negative effect of VPD on GPP, which is worthy to know because future projections of the land carbon sink depend on how models capture the response of GPP to atmospheric and soil droughts.

Thus, the work findings are quite significant to research field. Additionally, the paper is well-written and clear.

However, my main concern is about the lack of data uncertainty estimates and considerations about them. Could authors estimate or comment the input data uncertainty effects in the analyses of GPP, Gc, Amax and Vcmax sensitivities to SWC and VPD? In this sense, I also include two specific examples:

- Line 263-264: Were data uncertainties estimated? Include some details if so. Were accuracies took into account in the study? Were they used for establishing quality thresholds?
- Line 313, Eq. (3): taking into account the measurement uncertainties in the input data, could you provide an estimate of the Gc uncertainty?

Other specific comments:

- There are some text repetitions. Please, avoid them. E.g., line 195 and lines 185-186, but also lines 244-245.
- Line 210-211: Although something is mentioned in lines 93-94, explain briefly the selection of CMIP6 ESMs.
- Line 216-217: ESM underestimates should be quantified in the manuscript text (both here and in the conclusions).
- Line 259: include the range of surface depth variation across sites.
- Line 260: the processing pipeline should be shortly summarized here.
- Line 269: "croplands were excluded": Have you estimated the possible effect of croplands and their behavior in the ESM global results (and underestimates)? Please, add any comment about it.
- Line 285, Eq. (2): Could you briefly summarize the GPP sensitivity to the rest of terms in the manuscript

text?

- Line 399-400: include here appropriate references about the ESMs under study.
- Line 389: "Liu and colleagues" should be better "Liu et al."

Reviewer #2 (Remarks to the Author):

The study uses eddy-flux data to evaluate how soil water content (SWC) and air VPD (i.e. soil and atmospheric drought) affect GPP and some of its mechanistic determinants (G_c , A_{max} , V_{cmax} , recomputed by inverting Penman-Monteith equations and the "classic biochemical model") over multiple sites and several years using datasets including the fluxnet2015 dataset and the recent extreme drought over Europe in 2018 available in ICOS dataset.

The study also proposes to analyze whether Land Surface Model are able to reproduce the response derived from eddy-flux data.

Eddy-Flux data analyzed through ANN highlight three main different patterns of response, some of which are already documented, while other are much less, particularly at such a global scale. Three main patterns are found including :

- A negative effect of VPD on GPP and G_c which holds for the full range of VPD explored. Such effects is consistent with leaf-level and plant hydraulic knowledge.
- A negative effect of SWC on flux (GPP and G_c) for a restricted range of low SWC values. Such effects is known and consistent with leaf-level data and soil and plant hydraulic theory (see below).
- A positive effect of decrease soil water content on GPP, A_{max} , V_{cmax} , (but not on G_c) fluxes for relatively high SWC values, which is more surprising.

The two first patterns can be reviewed together as they are linked to drought effects on stomatal conductance mostly. They relate to the relative influence of the SWC and VPD on ecosystem functioning which have been the topic of some debates recently (in particular in some large scale remote sensing study). It appears important to provide large scale empirical evidence for the dominant contribution of VPD over SWC, which the author's do convincingly. However, I would suggest the authors to complement their discussion on the general effect of drought by providing a few mechanistic explanations. For instance, plant hydraulic studies have shown that stomatal closure respond tightly and early to leaf/canopy water potential (Martin-StPaul et al 2017), thus, it makes sense that VPD trigger stomatal closure all over its range (particularly if the light effect has been discarded) – as it affect water potential directly through transpiration-- whereas SWC only acts at relatively low value (given the nonlinear relationship between SWC and soil and plant water potential). Note that the increase sensitivity of GPP to VPD at low SWC can also be linked to increase leaf temperature. In addition, in last section, where LSM ability to reproduce empirical patterns are diagnosed, it would be nice that the author encourage implementation of plant hydraulic scheme in LSM to properly account for both effects (VPD and SWC) through plant functioning.

Regarding the third pattern (the positive sensitivity of GPP, Amax, Vcmax to decrease SWC), which is much rarely reported, I am far less competent, but... The authors report a positive sensitivity of GPP to decrease SWC while Gc remains constant which means the increase WUEi is due to increase investment in leaf nitrogen and carboxylation capacity. In fact, in Temperate and Mediterranean system, which I know better, an increase in WUEi generally occurs through decrease gs and a photosynthesis relatively maintained.

However in the data reported in the paper from Fu et al (Figure 2), there is no decrease in Gc visible for the range of SWC percentiles where GPP, Amax and Vcmax are affected (~80% to 100%, Figure 2). Actually, this is consistent with the fact that decrease SWC around high values has negligible effect on water potential (as explained above). The pattern reported may exist but is likely not related to drought. Consequently, I find a bit awkward that most of the discussion from L159-L164 is focused on drought WUE and NUE.

However, L147-148, the authors provides a couple of words about alternative explanations for the lower photosynthesis under wet soil, related to the potential effects of water logging on oxygen supply or nitrogen leaching. But I do not find this fully convincing :

- decrease oxygen supply should affect PH, aquaporins synthesis and thus root hydraulic conductance, water flux and GC which is not the case here.
- Nitrogen leaching is common during rain period, but why would leached nitrogen become more available to plants after rain has stopped ? Due to change microbial/fungal activity/community ? This would deserved additional references.

Finally, could the pattern reported be simply due to a temporal correlation between decrease SWC and leaf maturation (and thus N investment) instead ?

Overall, the third pattern reported may exist, but it is unclear and insufficiently explained at this stage.

To conclude, I found the paper quite clear and pedagogic in the presentation of the results. Also, the results are appealing and it appears important to report global patterns of response of flux to drought, in particular if they are not properly reproduced by land surface model. However to strengthen the MS I would recommend :

- that the first two patterns of response are discussed in the light of plant hydraulics and plant hydraulic models ;
- to provide additional explanations for the third pattern (increase GPP with decrease SWC) as they are not fully convincing to me at this stage.

Details

L385 : drier

REFERENCES

Martin-StPaul, N., Delzon, S., Cochard, H., 2017. Plant resistance to drought depends on timely stomatal

closure. *Ecol. Lett.* 20, 1437–1447. <https://doi.org/10.1111/ele.12851>

Dreyer, E., Colin-Belgrand, M., Biron, P., 1991. Photosynthesis and shoot water status of seedlings from different oak species submitted to waterlogging. *Ann. des Sci. For.* 48, 205–214.
<https://doi.org/10.1051/forest:19910207>

Rasheed-Depardieu, C., Parelle, J., Tatin-Froux, F., Parent, C., Capelli, N., 2015. Short-term response to waterlogging in *Quercus petraea* and *Quercus robur*: A study of the root hydraulic responses and the transcriptional pattern of aquaporins. *Plant Physiol. Biochem.* 97, 323–330.
<https://doi.org/10.1016/j.plaphy.2015.10.016>

**Atmospheric dryness reduces photosynthesis along a large range of soil water deficits
NCOMMS-21-22710-T**

Response to Reviewers

We greatly appreciate the constructive comments from the reviewers and the invitation from the editor to submit a revised version. We have thoroughly revised the manuscript following the reviewers' suggestions. Please see below our point-to-point responses in blue text following reviewer comments. The line numbers referred to are for the clean version of the revised manuscript (non-track-change version).

Reviewer #1 (Remarks to the Author):

1.1 The paper assessed gross primary production (GPP) sensitivities to low soil water content (SWC) and water vapor pressure deficit (VPD) to evaluate under what SWC and VPD conditions is GPP most negatively affected by droughts. Authors found that VPD dominated dryness stress on ecosystem production while SWC becomes important only under very dry soils. Their findings in these sense, after using three different procedures (linear, nonlinear model and the approach of ref.3), differ from a recent published paper (ref.3; also in Nat. Commun., 2020) that showed a global assessment of a predominant SWC stress on GPP using solar-induced chlorophyll fluorescence satellite observations and re-analysis data. After investigating the discrepancy, authors suggested to reexamine the role of VPD in studies that neglected the strong VPD-radiation coupling.

Another important finding is that 5 CMIP6 Earth System Models (ESM) underestimate the negative effect of VPD on GPP, which is worthy to know because future projections of the land carbon sink depend on how models capture the response of GPP to atmospheric and soil droughts.

Thus, the work findings are quite significant to research field. Additionally, the paper is well-written and clear.

We thank the reviewer for the very positive evaluation of the manuscript.

1.2 However, my main concern is about the lack of data uncertainty estimates and considerations about them. Could authors estimate or comment the input data uncertainty effects in the analyses of GPP, G_c , A_{max} and V_{cmax} sensitivities to SWC and VPD? In this sense, I also include two specific examples:

- Line 263-264: Were data uncertainties estimated? Include some details if so. Were accuracies took into account in the study? Were they used for establishing quality thresholds?
- Line 313, Eq. (3): taking into account the measurement uncertainties in the input data, could you provide an estimate of the G_c uncertainty?

Thank you for raising this important point. In this study, we used GPP from the nighttime

partitioning method; it was used as the benchmark for model validation¹, remote sensing products evaluation^{2,3} and many other studies^{1,4,5,6,7}. All variables in Eq. (3) and Eq. (6) deriving G_c and A_{max} were either measured directly or constants which are well known^{8,9,10}.

The GPP uncertainty mainly arises from net ecosystem CO₂ exchange (NEE) processing and flux partitioning methods¹¹. For the uncertainty of NEE processing, 25 sites provided the standard error of GPP (half-hourly) to quantify this uncertainty. We checked the sites that reported standard error of GPP and found that the standard errors were very small relative to GPP values (Fig. 1), thus this uncertainty should not affect our results because we used the GPP anomalies (z-scores).

Fig. 1 Examples at four sites (Supplementary Table 2) reported the standard error of GPP (half-hourly). We checked each site that reported standard error of GPP and found that the standard errors were very small relative to their GPP values for all these sites.

The measured NEE using the eddy covariance technique was partitioned into GPP and ecosystem respiration (RECO) using two different methods; the first is a nighttime based approach (NT, Reichstein, Falge¹²) where the nighttime data are used to parameterize a respiration model that is then applied to the whole dataset to estimate RECO. GPP is then calculated as difference between RECO and NEE. The second method is based on daytime data (DT, Lasslop, Reichstein¹³) that are used to parameterize a model where NEE is function of both GPP and RECO.

For the uncertainty of partitioning methods, we performed the same sensitivity analysis using GPP from the daytime partitioning method (GPP_DT)¹³, and compared the results obtained in our main analysis using GPP from the nighttime partitioning method (GPP_NT)¹², to estimate a systematic uncertainty of GPP sensitivity to SWC and VPD (Fig. 2). The patterns of GPP

sensitivity to SWC and VPD were found to be consistent between methods (Figs. 2a-d). The uncertainty from these two different partitioning methods were quantified by calculating the differences of their sensitivities (e.g., GPP_NT sensitivity to SWC minus GPP_DT sensitivity to SWC; or GPP_NT sensitivity to VPD minus GPP_DT sensitivity to VPD) for each bin (Eq.1, Figs. 2e-f). We also calculated the relative uncertainty using the absolute value of differences of sensitivities divided by the absolute value of mean sensitivities, indicating that the uncertainty represents how many percent of the mean sensitivity (Eq.2, Figs. 2g-h).

$$Uncertainty_{SWC} = Sensitivity_{GPP_NT\ to\ SWC} - Sensitivity_{GPP_DT\ to\ SWC} \quad (1)$$

$$Relative\ uncertainty_{SWC} = 100\% \times \left| \frac{Uncertainty_{SWC}}{(Sensitivity_{GPP_NT\ to\ SWC} + Sensitivity_{GPP_DT\ to\ SWC})/2} \right| \quad (2)$$

Across all bins, we found that the differences of GPP sensitivity values obtained between the two GPPs were falling mostly in the range -0.1 to 0.1 (Figs. 2e-f), and relative uncertainties in most bins were less than 20% (Figs. 2g-h), indicating the uncertainty of flux partitioning methods had minor effects on our results. Please note that the high levels of relative uncertainty occurred in the bins with statistically insignificant sensitivity values (Figs. 2a, c, g). Since these sensitivity values are close to zero, a low absolute uncertainty level leads to a high relative uncertainty level.

Fig. 2 Effect of the partitioning methods (Nighttime vs. Daytime) on GPP sensitivity to SWC and VPD. a-b, The sensitivity of GPP to SWC (a) and VPD (b) using GPP from nighttime partitioning method (GPP_NT). c-d, The sensitivity of GPP to SWC (c) and VPD (d) using GPP from daytime partitioning method (GPP_DT). e-f, Uncertainty in GPP sensitivity to SWC (e) and VPD (f). g-h, Relative uncertainty in GPP sensitivity to SWC (g) and VPD (h). Please note that the high levels of relative uncertainty occurred in the bins with statistically insignificant sensitivity values (a, c, g). Since these sensitivity values are close to zero, a low absolute uncertainty level leads to a high relative uncertainty level.

G_c , calculated using eddy covariance observations by inverting the Penman–Monteith equation, have been widely applied in previous studies^{9, 14, 15, 16, 17}. The main sources of uncertainty for G_c is the latent heat flux uncertainty from eddy covariance measurements. Both the ‘LE’ and

‘LE.CORR’ variables were reported by the ICOS and FLUXNET2015 database for latent energy exchange. LE.CORR is the “energy balance corrected” version of latent heat flux, based on the assumption that Bowen ratio is correct. We repeated our analysis using LE.CORR, comparing with the results obtained using LE, to estimate a systematic uncertainty of G_c sensitivities to SWC and VPD (Fig. 3). Our results were robust to either variable (Figs. 3a-d). The differences of G_c sensitivity values obtained between the two latent heat fluxes were falling mostly in the range -0.1 to 0.1 (Figs. 3e-f), and the relative uncertainties in most bins were less than 20% (Figs. 3g-h). Please note that the high levels of relative uncertainty occurred in the bins with statistically insignificant sensitivity values (Figs. 3a, c, g). Since these sensitivity values are close to zero, a low absolute uncertainty level leads to a high relative uncertainty level.

Fig. 3 Effect of different latent heat fluxes (LE vs. LE.CORR) on G_c sensitivity to SWC and VPD. a-b, The sensitivity of G_c to SWC (a) and VPD (b) using LE (G_c_LE). c-d, The sensitivity of G_c to SWC (c) and VPD (d) using LE.CORR ($G_c_LE.CORR$). e-f, Uncertainty in G_c sensitivity to SWC (e) and VPD (f). g-h, Relative uncertainty in G_c sensitivity to SWC (g) and VPD (h). Please note that the high levels of relative uncertainty occurred in the bins with statistically insignificant sensitivity values (a, c, g). Since these sensitivity values are close to zero, a low absolute uncertainty level leads to a high relative uncertainty level.

The uncertainty of A_{max} derived from light response curve using eddy covariance observations has been evaluated by Luo and Keenan¹⁰. They derived and compared the A_{max} and A_{2000} (ecosystem photosynthesis at a photosynthetic photon flux density of $2000 \mu\text{mol m}^{-2} \text{s}^{-1}$) from eddy covariance observations, and found that there was a very good consistency between A_{max} and A_{2000} (Fig. 4), indicating that the uncertainty in A_{max} from this method is small. Therefore, the uncertainty of A_{max} did not affect A_{max} sensitivity to SWC and VPD because the A_{max} anomalies (z-scores) were used in this analysis.

Fig. 4 Comparison between A_{\max} and A_{2000} (ecosystem photosynthesis at a photosynthetic photon flux density of $2000 \mu\text{mol m}^{-2} \text{s}^{-1}$)¹⁰. (This figure is a reproduction of Extended Data Figure 2 Panel A from Luo, X., Keenan, T.F. Global evidence for the acclimation of ecosystem photosynthesis to light. *Nat Ecol Evol* 4, 1351–1357 (2020). <https://doi.org/10.1038/s41559-020-1258-7>)

The effects of measurement uncertainties on V_{cmax} are difficult to assess because of a lack of repetition of the measurements of the variables used to derive V_{cmax} . However, this source of uncertainty should not hamper our results because the measurements were done at high frequency (half-hourly) and were automatic for all flux towers, thus with random errors mainly, and limiting the risk of bias.

Additionally, to further consider and estimate the uncertainties of our results as the reviewer suggested, we also quantified the uncertainties of GPP, G_c , A_{\max} and V_{cmax} sensitivities to SWC and VPD, respectively, by calculating the standard errors of sensitivities for each bin across all sites (Fig. 5). We found that in most bins, the standard errors of GPP, G_c , A_{\max} and V_{cmax} sensitivities to SWC or VPD were less than 0.1 (Fig. 5). In a few cases, higher standard errors (0.15-0.2) were found, mainly in the bins with both high SWC and high VPD (80-100th percentile, Figs. 5a, b, c, e, f, h) or both low SWC and low VPD simultaneously (0-20th percentile, Figs. 5c, d, e, g). This is because the number of data points were less in these ‘extreme’ bins (Supplementary Fig. 4 in the revised manuscript). We also calculated the relative uncertainties of GPP, G_c , A_{\max} and V_{cmax} sensitivities to SWC and VPD, respectively, using the standard errors divided by the absolute value of median sensitivities (Fig. 6). The relative uncertainties of GPP, G_c , A_{\max} and V_{cmax} sensitivities to SWC in most bins were less than 20% (Figs. 6a, c, e, g) while relative uncertainties to VPD in most bins were less than 10% (Figs. 6b, d, f, h). Please note that the high levels of relative uncertainty occurred in the bins with statistically insignificant sensitivity values (Figs. 6a, c, e, g below and Fig. 2 in the revised manuscript). Since these sensitivity values are close to zero, a low absolute uncertainty level leads to a high relative uncertainty level.

Fig. 5 Standard error of the sensitivity of GPP (a-b), G_c (c-d), A_{max} (e-f), and V_{cmax} (g-h) to SWC (a, c, e, g) and VPD (b, d, f, h) for each bin across all sites.

Fig. 6 Relative uncertainty of the sensitivity of GPP (a-b), G_c (c-d), A_{max} (e-f), and V_{cmax} (g-h) to SWC (a, c, e, g) and VPD (b, d, f, h) for each bin. Please note that the high levels of relative uncertainty occurred in the bins with statistically insignificant sensitivity values (a, c, e, g). Since these sensitivity values are close to zero (Fig. 2 in the revised manuscript), a low absolute uncertainty level leads to a high relative uncertainty level.

Following to the reviewer's suggestions, we have estimated and described the uncertainties of GPP, G_c , A_{max} and V_{cmax} sensitivities to SWC and VPD both in Results (Line 116-126; Line 189-195) and Methods (Line 423-456).

Line 116-126 (Results section):

“Uncertainty in GPP data mainly arises from net ecosystem CO_2 exchange (NEE) processing and flux partitioning methods¹¹. For the uncertainty of NEE processing, some sites provided the

standard error of GPP to quantify the uncertainty. We checked the sites that reported standard error of GPP and found that the standard errors were very small relative to GPP values (Supplementary Fig. 6), thus this uncertainty should not affect our results because we used the GPP anomalies (z-scores). Concerning partitioning methods, we repeated our analysis using GPP from the daytime partitioning method¹³, and compared the results obtained in our main analysis using GPP from the nighttime partitioning method¹² (Methods). The patterns of GPP sensitivity to SWC and VPD were found to be consistent between methods (Supplementary Figs. 7a-d). Across all bins, the differences in GPP sensitivity values based on the two partitioning methods mostly fell in the range from -0.1 to 0.1 (Supplementary Figs. 7e-f), indicating that flux partitioning uncertainties had minor effects on our results. ”

Line 189-195 (Results section):

“To further consider and estimate the uncertainties of our results, we quantified the uncertainties of GPP, G_c , A_{max} and V_{cmax} sensitivities to SWC and VPD, respectively, by calculating the standard errors of sensitivities for each bin across all sites (Methods). We found that in most bins, the standard errors of GPP, G_c , A_{max} and V_{cmax} sensitivities to SWC or VPD were less than 0.1 (Supplementary Fig. 11). In a few cases, standard errors were higher (0.15-0.2), mainly in the bins with simultaneously high or low SWC and VPD (80-100th percentiles or 0-20th percentiles, Supplementary Fig. 11), where there were fewer data points (Supplementary Fig. 4). ”

Line 423-456 (Methods section):

“The uncertainty of GPP used in this study mainly arises from net ecosystem CO₂ exchange (NEE) processing and flux partitioning methods¹¹. Concerning NEE processing, some sites that reported the standard error of GPP to quantify the uncertainty were checked. For the uncertainty of partitioning methods, we repeated the sensitivity analysis using GPP from the daytime partitioning method (GPP_DT)¹³, and compared the results obtained in our main analysis using GPP from the nighttime partitioning method (GPP_NT)¹². The uncertainty from these two different partitioning methods were quantified by calculating the differences of their sensitivities (e.g., GPP_NT sensitivity to SWC minus GPP_DT sensitivity to SWC) for each bin (Eq. 11, Supplementary Figs. 7e-f). We also calculated the relative uncertainty using the absolute value of differences of sensitivities divided by the absolute value of mean sensitivities, indicating that the uncertainty represents how many percent of the mean sensitivity (Eq. 12, Supplementary Figs. 7g-h). Please note that the high levels of relative uncertainty occurred in the bins with statistically insignificant sensitivity values (Supplementary Figs. 7a, c, g). Since these sensitivity values are close to zero, a low absolute uncertainty level leads to a high relative uncertainty level.

$$Uncertainty_{SWC} = Sensitivity_{GPP_NT\ to\ SWC} - Sensitivity_{GPP_DT\ to\ SWC} \quad (11)$$

$$Relative\ uncertainty_{SWC} = 100\% \times \left| \frac{Uncertainty_{SWC}}{(Sensitivity_{GPP_NT\ to\ SWC} + Sensitivity_{GPP_DT\ to\ SWC})/2} \right| \quad (12)$$

The main sources of uncertainty for G_c is the latent heat flux uncertainty from eddy covariance measurements. We used both the ‘LE’ and ‘LE.CORR’ variables reported by the ICOS and FLUXNET2015 database for latent energy exchange. LE.CORR is the “energy balance corrected” version of latent heat flux, based on the assumption that Bowen ratio is correct. Our results were robust to either variable (Supplementary Fig. 13). The differences in G_c sensitivity

values based on the two latent heat fluxes mostly fell in the range from -0.1 to 0.1 (Supplementary Figs. 13e-f). The uncertainty of A_{\max} was evaluated by Luo and Keenan¹⁰, who showed that the values of A_{\max} and A_{2000} (ecosystem photosynthesis at a photosynthetic photon flux density of $2000 \mu\text{mol m}^{-2} \text{s}^{-1}$) were very consistent, indicating that the uncertainty in A_{\max} from this method is small. The effects of measurement uncertainties on V_{cmax} are difficult to assess because of a lack of repetition of the measurements of the variables used to derive V_{cmax} . However, this source of uncertainty should not hamper our results because the measurements were done at high frequency and were automatic for all flux towers, thus with random errors mainly, and limiting the risk of bias.

Additionally, to further consider and estimate the uncertainties of our results, we quantified the uncertainties of GPP, G_c , A_{\max} and V_{cmax} sensitivities to SWC and VPD, respectively, by calculating the standard errors of sensitivities for each bin across all sites (Supplementary Fig. 11). We also calculated the relative uncertainties of GPP, G_c , A_{\max} and V_{cmax} sensitivities to SWC and VPD, respectively, using the standard errors divided by the absolute value of median sensitivities (Supplementary Fig. 12). ”

Other specific comments:

1.3 - There are some text repetitions. Please, avoid them. E.g., line 195 and lines 185-186, but also lines 244-245.

Thanks for pointing this out. We deleted the text in Line 195 and Lines 244-245. We also double checked this through the manuscript.

1.4 Line 210-211: Although something is mentioned in lines 93-94, explain briefly the selection of CMIP6 ESMs.

Thank you for the suggestion. We modified this sentence to : “Last, we diagnosed the sensitivities of daily GPP to SWC and VPD from five CMIP6 ESMs (ACCESS-ESM1-5, CMCC-CM2-SR5, IPSL-CM6A-LR, NorESM2-LM and NorESM2-MM, Supplementary Table. 3), all of which provided daily outputs. ”

1.5 Line 216-217: ESM underestimates should be quantified in the manuscript text (both here and in the conclusions).

Thank you for this suggestion. In the revised manuscript, we have stated the values explicitly both in results (Line 243-245) and conclusions (Line 274-276) as suggested.

Line 243-245:

“However, all ESMs underestimated both the negative sensitivity of GPP to increasing VPD (0.19 ± 0.12 , median across five ESMs \pm standard error) and its positive sensitivity to decreasing SWC (-0.20 ± 0.07) at high SWC (> 80 th percentiles) (Figs. 4k-t).”

Line 274-276:

“The five state-of-the-art ESMs that we tested failed to accurately reproduce the magnitudes of sensitivities, underestimating them by about 0.2 for both the negative sensitivity to VPD and the positive sensitivity to SWC at high SWC levels. ”

1.6 Line 259: include the range of surface depth variation across sites.

We added the range of surface depth variation as the reviewer suggested (Line 287-288).

Line 287-288:

“SWC was measured as volumetric soil water content (percentage) at a surface depth (varying across sites, 5-30 cm).”

1.7 Line 260: the processing pipeline should be shortly summarized here.

Thank you for this suggestion. We have added the following statement to the revised manuscript.

Line 289-293:

“The data processing pipeline mainly included: (1) thorough data quality control checks; (2) calculation of a range of friction velocity thresholds; (3) gap-filling of meteorological and flux measurements; (4) partitioning of CO₂ fluxes into respiration and photosynthesis components; and (5) calculation of a correction factor for energy fluxes¹⁸. ”

1.8 Line 269: “croplands were excluded”: Have you estimated the possible effect of croplands and their behavior in the ESM global results (and underestimates)? Please, add any comment about it.

Thank you for pointing this out. Croplands were excluded due to the effect of management on the seasonal timing of ecosystem fluxes, both from crop rotation that change from year to year and from the variable timing of planting and harvesting. For ESMs results, we only extracted the output variables from each model according to the flux tower site locations (no cropland sites). All extracted pixels were not dominated by croplands. We didn't estimate the possible effect of croplands because croplands were not included in this study.

Following to the reviewer's suggestions, we added the description about the possible effect of croplands in Line 301-307:

“Croplands were excluded due to the effect of management on the seasonal timing of ecosystem fluxes, both from crop rotation that change from year to year and from the variable timing of planting and harvesting. In croplands, the changes of GPP anomalies across different growing season could be mainly depend on crop varieties and management activities. Information of crop varieties, growing times yearly and other management data for each cropland site should be collected in future in order to fully consider and disentangle the impacts of SWC and VPD on its photosynthesis. ”

1.9 Line 285, Eq. (2): Could you briefly summarize the GPP sensitivity to the rest of terms in the manuscript text?

Thank you for this suggestion. We have summarized the GPP sensitivity to air temperature and incoming shortwave radiation in Line 94-96:

“The positive linear regression slope between GPP and air temperature was slightly lower in

2014–2018 than in 2014–2017 while they were similar for incoming shortwave radiation (Supplementary Fig. 2).

Supplementary Fig. 2 Probability distributions across all sites for the linear regression slope of daily GPP anomalies to air temperature (T_a , a) and incoming shortwave radiation anomalies (RAD, b) during the summer across 2014–2018 and 2014–2017. ”

1.10 Line 399-400: include here appropriate references about the ESMs under study.

Thank you for this suggestion. We have added appropriate references for each ESM as the reviewer suggested (Line 469-471).

Line 469-471:

“Five Earth system models (ACCESS-ESM1-5¹⁹, CMCC-CM2-SR5²⁰, IPSL-CM6A-LR²¹, NorESM2-LM²² and NorESM2-MM²²) in CMIP6 provide daily GPP; most models provide only monthly GPP outputs (Supplementary Table 3). ”

1.11 Line 389: “Liu and colleagues” should be better “Liu et al.”

Changed as the reviewer suggested.

Reviewer #2 (Remarks to the Author):

2.1 The study uses eddy-flux data to evaluate how soil water content (SWC) and air VPD (i.e. soil and atmospheric drought) affect GPP and some of its mechanistic determinants (G_c , A_{max} , V_{cmax} , recomputed by inverting Penman-Monteith equations and the “classic biochemical model”) over multiple sites and several years using datasets including the fuxnet2015 dataset and the recent extreme drought over Europe in 2018 available in ICOS dataset.

The study also proposes to analyze whether Land Surface Model are able to reproduce the response derived from eddy-flux data.

Thank you for reviewing our paper and for your very helpful suggestions.

2.2 Eddy-Flux data analyzed through ANN highlight three main different patterns of response, some of which are already documented, while other are much less, particularly at such a global scale. Three main patterns are found including :

- A negative effect of VPD on GPP and G_c which holds for the full range of VPD explored. Such effects is consistent with leaf-level and plant hydraulic knowledge.
- A negative effect of SWC on flux (GPP and G_c) for a restricted range of low SWC values. Such effects is known and consistent with leaf-level data and soil and plant hydraulic theory (see below).
- A positive effect of decrease soil water content on GPP, A_{max} , V_{cmax} , (but not on G_c) fluxes for relatively high SWC values, which is more surprising.

The two first patterns can be reviewed together as they are linked to drought effects on stomatal conductance mostly. They relate to the relative influence of the SWC and VPD on ecosystem functioning which have been the topic of some debates recently (in particular in some large scale remote sensing study). It appears important to provide large scale empirical evidence for the dominant contribution of VPD over SWC, which the author's do convincingly. However, I would suggest the authors to complement their discussion on the general effect of drought by providing a few mechanistic explanations. For instance, plant hydraulic studies have shown that stomatal closure respond tightly and early to leaf/canopy water potential (Martin-StPaul et al 2017), thus, it makes sense that VPD trigger stomatal closure all over its range (particularly if the light effect has been discarded) – as it affect water potential directly through transpiration-- whereas SWC only acts at relatively low value (given the nonlinear relationship between SWC and soil and plant water potential). Note that the increase sensitivity of GPP to VPD at low SWC can also be linked to increase leaf temperature. In addition, in last section, where LSM ability to reproduce empirical patterns are diagnosed, it would be nice that the author encourage implementation of plant hydraulic scheme in LSM to properly account for both effects (VPD and SWC) through plant functioning.

We thank the reviewer very much for the thoughtful comments and valuable suggestions. As the reviewer suggested, we added the discussion on the general effect of drought by providing mechanistic explanations in Line 168-177. In addition, in the last section, we added the recommendation of implementing plant hydraulic schemes in ESMs (Line 254-256).

Line 168-177:

“Both GPP and G_c have negative sensitivity to increasing VPD across the full range of VPD and SWC (Figs. 2d-e) while their negative sensitivity to decreasing SWC mainly occurs in a restricted range of low SWC values (Figs. 2a-b). These results are consistent with stomatal closure responses documented at leaf scale, and with plant hydraulic theory^{23, 24}. Stomatal closure limits decreases in water potential in the plant, ensuring that water demand from the leaves does not exceed the supply capacity of the hydraulic system – which could lead to embolism of the vascular system and even, potentially, complete desiccation of the plant. Stomatal closure responds tightly and early to leaf/canopy water potential²⁴, thus increasing VPD triggers stomatal closure for the full range – as it affect water potential directly through transpiration. However, SWC only acts at relatively low values, i.e., below a threshold, given the nonlinear relationship between SWC and soil and plant water potential²³. ”

Line 254-256:

“Our results also suggest that the implementation of plant hydraulics in ESMs should allow both effects (VPD and SWC) to be better represented, because plant hydraulics play a critical role in leaf gas exchange by regulating stomatal conductance. ”

2.3 Regarding the third pattern (the positive sensitivity of GPP, A_{max} , V_{cmax} to decrease SWC), which is much rarely reported, I am far less competent, but... The authors report a positive sensitivity of GPP to decrease SWC while G_c remains constant which means the increase WUEi is due to increase investment in leaf nitrogen and carboxylation capacity. In fact, in Temperate and Mediterranean system, which I know better, an increase in WUEi generally occurs through decrease g_s and a photosynthesis relatively maintained. However in the data reported in the paper from Fu et al (Figure 2), there is no decrease in G_c visible for the range of SWC percentiles where GPP, A_{max} and V_{cmax} are affected (~80% to 100%, Figure 2). Actually, this is consistent with the fact that decrease SWC around high values has negligible effect on water potential (as explained above). The pattern reported may exist but is likely not related to drought. Consequently, I find a bit awkward that most of the discussion from L159-L164 is focused on drought WUE and NUE.

Thank you for this suggestion. We deleted the text in Line 159-164 and added the discussion on water potential as the reviewer suggested (Line 161-167).

Line 161-167:

“We also noted that the sensitivity of G_c to decreasing SWC under wet soil conditions is insignificant (Figs. 2c). There are two possible reasons for this. First, decreasing SWC in wet soils has negligible effect on canopy water potential so that there is no significant impact on G_c ²³. Second, species-specific effects may be involved. For example, Rasheed-Depardieu, Parelle²⁵ reported that waterlogging in *Quercus petraea* and *Quercus robur* decreased stomatal conductance while Yordanova and Popova²⁶ showed that there were no significant changes in stomatal conductance for maize plants. ”

2.4 However, L147-148, the authors provides a couple of words about alternative explanations for the lower photosynthesis under wet soil, related to the potential effects of water logging on oxygen supply or nitrogen leaching. But I do not find this fully convincing :

- decrease oxygen supply should affect PH, aquaporins synthesis and thus root hydraulic

conductance, water flux and GC which is not the case here.

- Nitrogen leaching is common during rain period, but why would leached nitrogen become more available to plants after rain has stopped ? Due to change microbial/fungal activity/community ? This would deserved additional references.

Finally, could the pattern reported be simply due to a temporal correlation between decrease SWC and leaf maturation (and thus N investment) instead ?

Overall, the third pattern reported may exist, but it is unclear and insufficiently explained at this stage.

We thank the reviewer for this important point. Yes, we found that when soils are wet, moderate soil drying increases GPP and A_{max} , but has no significant effect on G_c across all sites (Figs. 2b-c in the manuscript), which may result from two reasons. First, decreasing SWC in wet soils has negligible effect on canopy water potential so that there is no significant impact on G_c ²³. Second, the insignificant sensitivity of G_c to SWC might be also caused by species-specific effects. Experimental studies at the species level have documented that waterlogging decreased the rate of photosynthesis^{25, 27}, the activity of Rubisco²⁶, and chlorophyll fluorescence²⁸. But for stomatal conductance, the experimental results differed among different species. For example, Rasheed-Depardieu, Parelle²⁵ reported that waterlogging in *Quercus petraea* and *Quercus robur* decreased stomatal conductance while Yordanova and Popova²⁶ showed that there were no significant changes in stomatal conductance for maize plants. Our analysis broken down by different plant functional types found that broadleaved deciduous forests and evergreen needle-leaved forests tend to increase G_c in response to decreasing SWC under wet soils while grasslands and savannas tend to decrease G_c (Supplementary Fig. 9). This might be because of the difference in hydraulic strategy (e.g., isohydric vs. anisohydric plants) or rooting depth for different species^{23, 29}.

We fully agree that nitrogen leaching is common during rain period, but please note that our results are comparing the nitrogen contents between before rainfall and after rainfall, rather than the comparison between during rainfall and after. Waterlogging could decrease nitrogen availability due to leaching or denitrification, and increase exposure to toxic compounds and disease organisms^{30, 31, 32}, reducing photosynthesis. Meta-analysis also documented that moderate soil drying increased foliar and root nitrogen concentrations, with upregulation of root primary metabolism³³. Therefore, moderate soil drying of wet soils might help increase nitrogen uptake from the soil, thus enhance the A_{max} and GPP.

We also looked at the temporal change of daily GPP and SWC (e.g, Fig. 7 at FR-LBr site), and found that the this pattern (increasing GPP with decreasing SWC) reported is not simply due to a temporal correlation between decreasing SWC and leaf maturation.

Fig. 7 Example of time series showing the temporal change of daily GPP and SWC for the years available (2005-2008) at the site FR-LBr (Supplementary Table 2). 30-days moving averages were also plotted to smooth the day-to-day variation.

In the revised manuscript, we have rewritten this part and provided additional explanations for the positive sensitivity of GPP and A_{max} to decreasing SWC under wet soils according to the reviewer's suggestions (Line 145-152).

Line 145-152:

“The above analysis shows that when soils are wet, moderate soil drying is in fact accompanied by an increase in GPP. Indeed, moderate soil drying of wet soils might help increase photosynthetic biochemical activity and nitrogen uptake³². Experimental studies at the species level have documented that waterlogging decreased the rate of photosynthesis^{25, 27}, the activity of Rubisco²⁶, and chlorophyll fluorescence²⁸. Waterlogging could also decrease nitrogen availability due to leaching or denitrification, and increase exposure to toxic compounds and disease organisms^{30, 31, 32}, reducing photosynthesis. Global meta-analysis has also shown that moderate soil drying increases foliar and root nitrogen concentrations, with upregulation of root primary metabolism³³. ”

2.5 To conclude, I found the paper quite clear and pedagogic in the presentation of the results. Also, the results are appealing and it appears important to report global patterns of response of flux to drought, in particular if they are not properly reproduced by land surface model. However to strengthen the MS I would recommend :

- that the first two patterns of response are discussed in the light of plant hydraulics and plant hydraulic models ;
- to provide additional explanations for the third pattern (increase GPP with decrease SWC) as they are not fully convincing to me at this stage.

We greatly appreciate the reviewer's positive comments and support of the manuscript. We have carefully followed the reviewer's suggestions and revised the manuscript as above. As a consequence, our manuscript has been considerably improved. Please see our detailed responses above.

2.6 Details
L385 : drier

We corrected it, thank you for the careful read.

REFERENCES

Martin-StPaul, N., Delzon, S., Cochard, H., 2017. Plant resistance to drought depends on timely stomatal closure. *Ecol. Lett.* 20, 1437–1447. <https://doi.org/10.1111/ele.12851>

Dreyer, E., Colin-Belgrand, M., Biron, P., 1991. Photosynthesis and shoot water status of seedlings from different oak species submitted to waterlogging. *Ann. des Sci. For.* 48, 205–214. <https://doi.org/10.1051/forest:19910207>

Rasheed-Depardieu, C., Parelle, J., Tatin-Froux, F., Parent, C., Capelli, N., 2015. Short-term response to waterlogging in *Quercus petraea* and *Quercus robur*: A study of the root hydraulic responses and the transcriptional pattern of aquaporins. *Plant Physiol. Biochem.* 97, 323–330. <https://doi.org/10.1016/j.plaphy.2015.10.016>

References mentioned in the responses

1. Slevin D, Tett S, Williams M. Multi-site evaluation of the JULES land surface model using global and local data. *Geoscientific Model Development* **8**, 295-316 (2015).
2. Li X, *et al.* Solar-induced chlorophyll fluorescence is strongly correlated with terrestrial photosynthesis for a wide variety of biomes: First global analysis based on OCO-2 and flux tower observations. *Global change biology* **24**, 3990-4008 (2018).
3. Zhang Y, Joiner J, Alemohammad SH, Zhou S, Gentine P. A global spatially contiguous solar-induced fluorescence (CSIF) dataset using neural networks. *Biogeosciences* **15**, 5779-5800 (2018).
4. Wolf S, *et al.* Warm spring reduced carbon cycle impact of the 2012 US summer drought. *Proceedings of the National Academy of Sciences of the United States of America* **113**, 5880-5885 (2016).
5. Zhou S, Zhang Y, Williams AP, Gentine P. Projected increases in intensity, frequency, and terrestrial carbon costs of compound drought and aridity events. *Science advances* **5**, eaau5740 (2019).
6. Xia J, *et al.* Joint control of terrestrial gross primary productivity by plant phenology and physiology. *Proceedings of the National Academy of Sciences* **112**, 2788-2793 (2015).
7. Keenan TF, *et al.* Net carbon uptake has increased through warming-induced changes in temperate forest phenology. *Nature Climate Change* **4**, 598-604 (2014).
8. Allen RG, Pereira LS, Raes D, Smith M. Crop evapotranspiration-Guidelines for computing crop water requirements-FAO Irrigation and drainage paper 56. *Fao, Rome* **300**, D05109 (1998).
9. Kimm H, *et al.* Redefining droughts for the U.S. Corn Belt: The dominant role of atmospheric vapor pressure deficit over soil moisture in regulating stomatal behavior of Maize and Soybean. *Agricultural and Forest Meteorology* **287**, 107930 (2020).
10. Luo X, Keenan TF. Global evidence for the acclimation of ecosystem photosynthesis to light. *Nat Ecol Evol*, (2020).
11. Papale D, *et al.* Towards a standardized processing of Net Ecosystem Exchange measured with eddy covariance technique: algorithms and uncertainty estimation. *Biogeosciences* **3**, 571-583 (2006).
12. Reichstein M, *et al.* On the separation of net ecosystem exchange into assimilation and ecosystem respiration: review and improved algorithm. *Global Change Biology* **11**, 1424-1439 (2005).

13. Lasslop G, *et al.* Separation of net ecosystem exchange into assimilation and respiration using a light response curve approach: critical issues and global evaluation. *Global Change Biology* **16**, 187-208 (2010).
14. Novick KA, *et al.* The increasing importance of atmospheric demand for ecosystem water and carbon fluxes. *Nature Climate Change* **6**, 1023 (2016).
15. Bai Y, *et al.* Quantifying plant transpiration and canopy conductance using eddy flux data: An underlying water use efficiency method. *Agricultural and Forest Meteorology* **271**, 375-384 (2019).
16. Green J, Berry J, Ciais P, Zhang Y, Gentine P. Amazon rainforest photosynthesis increases in response to atmospheric dryness. *Science Advances* **6**, eabb7232 (2020).
17. Lin C, Gentine P, Huang Y, Guan K, Kimm H, Zhou S. Diel ecosystem conductance response to vapor pressure deficit is suboptimal and independent of soil moisture. *Agricultural and Forest Meteorology* **250**, 24-34 (2018).
18. Pastorello G, *et al.* The FLUXNET2015 dataset and the ONEFlux processing pipeline for eddy covariance data. *Scientific data* **7**, 1-27 (2020).
19. Ziehn T, *et al.* The Australian Earth System Model: ACCESS-ESM1. 5. *Journal of Southern Hemisphere Earth Systems Science*, (2020).
20. Cherchi A, *et al.* Global Mean Climate and Main Patterns of Variability in the CMCC-CM2 Coupled Model. *Journal of Advances in Modeling Earth Systems* **11**, 185-209 (2019).
21. Lurton T, *et al.* Implementation of the CMIP6 Forcing Data in the IPSL-CM6A-LR Model. *Journal of Advances in Modeling Earth Systems* **12**, e2019MS001940 (2020).
22. Seland Ø, *et al.* The Norwegian Earth System Model, NorESM2–Evaluation of the CMIP6 DECK and historical simulations. *Geoscientific Model Development Discussions*, 1-68 (2020).
23. Bonan G. *Climate change and terrestrial ecosystem modeling*. Cambridge University Press (2019).
24. Martin-StPaul N, Delzon S, Cochard H. Plant resistance to drought depends on timely stomatal closure. *Ecology Letters* **20**, 1437-1447 (2017).
25. Rasheed-Depardieu C, Parelle J, Tatin-Froux F, Parent C, Capelli N. Short-term response to waterlogging in *Quercus petraea* and *Quercus robur*: A study of the root hydraulic responses and the transcriptional pattern of aquaporins. *Plant Physiology and Biochemistry* **97**, 323-330 (2015).

26. Yordanova RY, Popova LP. Flooding-induced changes in photosynthesis and oxidative status in maize plants. *Acta Physiologiae Plantarum* **29**, 535-541 (2007).
27. Dreyer E, Colin-Belgrand M, Biron P. Photosynthesis and shoot water status of seedlings from different oak species submitted to waterlogging. In *Annales des sciences forestières* (Vol. 48, No. 2, pp. 205-214). EDP Sciences (1991).
28. Ren B, Zhang J, Dong S, Liu P, Zhao B. Effects of waterlogging on leaf mesophyll cell ultrastructure and photosynthetic characteristics of summer maize. *PloS one* **11**, e0161424 (2016).
29. Konings AG, Gentine P. Global variations in ecosystem-scale isohydricity. *Global Change Biology* **23**, 891-905 (2017).
30. Rigden A, Mueller N, Holbrook N, Pillai N, Huybers P. Combined influence of soil moisture and atmospheric evaporative demand is important for accurately predicting US maize yields. *Nature Food* **1**, 127-133 (2020).
31. Voesenek LA, Bailey-Serres J. Flood adaptive traits and processes: an overview. *New Phytologist* **206**, 57-73 (2015).
32. Li Y, Guan K, Schnitkey GD, DeLucia E, Peng B. Excessive rainfall leads to maize yield loss of a comparable magnitude to extreme drought in the United States. *Global Change Biology* **25**, 2325-2337 (2019).
33. Sardans J, *et al.* Changes in nutrient concentrations of leaves and roots in response to global change factors. *Global change biology* **23**, 3849-3856 (2017).

REVIEWER COMMENTS

Reviewer #1 (Remarks to the Author):

The paper has been remarkably improved after the revision taking into account the reviewers' comments. Thus, it can be accepted as it is.

Reviewer #3 (Remarks to the Author):

The paper is interesting and well designed and written. I entered in the revision process in the second submission and the changes following the two reviewers definitely improved the manuscript. I think however that one main aspect should be better addressed and a number of less crucial sentences or sections improvements are also possible.

My main concern is on a point raised by the Reviewer 2 where I don't think the answer is sufficient. It is related to the possible temporal correlation between SWC decrease and leaves maturation (but I would say more generally phenological cycle). I think this should be better evaluated and proved and I find the single site example plot (Figure 7, only in the answers to reviewers) not sufficient because only one example and because only visually interpreted. In addition, even based on this single plot, it looks like there some fundament in the issue raised: GPP and SWC have in general a two-picks pattern with the SWC reaching the maximum in early spring and then decreasing in the period when GPP increases. The spring increase of GPP is part of the phenology I would say and not driven by the SWC reduction (or t least not only). In fact if you look at 2005 or 2007 the increase of GPP is present even without a reduction of SWC (that in these two years is less then 50%).

The other observations I have are listed here below:

- 1) SWC is considered only until 30 cm and Water Table Depth has not been considered at all, but for trees these can be important and water access can happen even if the first layers of soil are dry. This should be at least discussed, better analysed if there are sites with these data.
- 2) Did you filtered and evaluated also the SWC and VPD quality? There are sites where SWC is in the first percentile (very dry) and VPD is very low and also sites with very high SWC and very high VPD. In some conditions this could be possible (the second case more difficult) but it is important to verify that it is not an artefact due to data quality issues.
- 3) The evaluation of the uncertainty in GPP was a good addition and it is convincing. It must be also noted (and may be reported) that for the estimation of GPP_DT the VPD is used as limiting factor so there is a risk of circularity. For this reason it was a good choice to use the GPP NT. I think that the Standard Error is however not the best choice for the uncertainty analysis because the uncertainty is not normally distributes. I would suggest to evaluate the uncertainty in the NT using the quartile ranges, always available for all the sites in both the collections used and derived by the uncertainty in NEE.
- 4) The reference to "extreme dry soil" (line 208) or SWC "extremely low" (line 214) is a little bit too strong in my opinion. The two effects (SWC and VPD) are basically equivalent from the 40th percentile

of SWC, that means almost half of the range of values... I would not consider this “extreme”. In general I found the overall message a little bit too strong in highlighting the role of VPD respect to SWC while the data presented show that when limiting the SWC has also a strong role. Don't you agree?

5) At line 289 it would be nice and useful to better clarify and report that all the corrections listed are already applied to the available product (so it has not been done by the authors for the specific study). This would promote the use of these open access data.

6) Line 338: it would help to refer to the following section (page 23) where the method is introduced and explained.

7) At lines 342-343 you talk about “binned SWC and VPD anomalies into 10x10 percentile bins”. In fact you didn't bin the anomalies but the SWC and VOD values (normalized, but I don't think you can call them anomalies, it is the natural variability). I also suggest that the caption of Figure 2 should explain what are the percentiles in order to make it more self-explaining.

8) On the Code availability it is stated that it is available upon request. I strongly suggest that at least the codes to calculate the G_c , A_{max} , and V_{cmax} should be openly shared. There could be (I'm sure) eddy covariance people that would like to use these codes and it is a good feedback toward people that shared the data for the paper. So please share the code in a repository and allow people to calculate the same parameters you estimated in the same way in order to do further analysis on the topic...

Dario Papale

**Atmospheric dryness reduces photosynthesis along a large range of soil water deficits
NCOMMS-21-22710A**

Response to Reviewers

We greatly appreciate the constructive comments from the reviewers and the invitation from the editor to submit a revised version. We have thoroughly revised the manuscript following the reviewers' suggestions. Please see below our point-to-point responses in blue text following reviewer comments. The line numbers referred to are for the clean version of the revised manuscript (non-track-change version).

Reviewer #1 (Remarks to the Author):

1.1 The paper has been remarkably improved after the revision taking into account the reviewers' comments. Thus, it can be accepted as it is.

We thank the reviewer for supporting this paper and for recognizing our effort to improve it.

Reviewer #3 (Remarks to the Author):

3.1 The paper is interesting and well designed and written. I entered in the revision process in the second submission and the changes following the two reviewers definitely improved the manuscript. I think however that one main aspect should be better addressed and a number of less crucial sentences or sections improvements are also possible.

We appreciate Dr. Papale's positive comments and very helpful suggestions. We have thoroughly revised the manuscript following your suggestions. Please see below our detailed responses.

3.2 My main concern is on a point raised by the Reviewer 2 where I don't think the answer is sufficient. It is related to the possible temporal correlation between SWC decrease and leaves maturation (but I would say more generally phenological cycle). I think this should be better evaluated and proved and I find the single site example plot (Figure 7, only in the answers to reviewers) not sufficient because only one example and because only visually interpreted. In addition, even based on this single plot, it looks like there some fundament in the issue raised: GPP and SWC have in general a two-picks pattern with the SWC reaching the maximum in early spring and then decreasing in the period when GPP increases. The spring increase of GPP is part of the phenology I would say and not driven by the SWC reduction (or at least not only). In fact if you look at 2005 or 2007 the increase of GPP is present even without a reduction of SWC (that in these two years is less then 50%).

Thank you very much for your suggestions. This study focused on the growing season and days when the SWC and VPD effects were most likely to control ecosystem fluxes and screen out days when other meteorological drivers were likely to have a larger influence on fluxes.

Following previous studies^{1,2,3}, for each site, we restrict our analyses to the days in which: (i) the daily average temperature >15 °C; (ii) sufficient evaporative demand existed to drive water fluxes, constrained as daily average VPD > 0.5 kPa; (iii) high solar radiation, constrained as daily average incoming shortwave radiation >250 Wm^{-2} . Thus, our analyses were mainly in days during the peak of growing season, therefore limiting the impact of phenology. We point this out in Line 332-337.

To further test if the phenological cycle affects our results, we repeated our analysis using only peak growing season, defined as 3-month period with the maximum mean GPP across the available years, where seasonal variability is muted. We found that the patterns of GPP sensitivity to SWC and VPD using only peak growing season were consistent with the results in our main analysis (Figs. 1a, b).

Additionally, we calculated the seasonal cycle in GPP by averaging all available years of the data and smoothing the series with a 30-day moving average as Feldman, Short Gianotti⁴, then we removed the seasonal cycle and repeated the sensitivity analysis. The results show that doing so does not change our conclusions (Figs. 1c, d).

Fig. 1 The sensitivity of GPP to SWC (a) and VPD (b) using only peak growing season, the 3-month period with the maximum mean GPP across the available years, where seasonal variability is muted. The sensitivity of GPP to SWC (c) and VPD (d) using anomalies by removing the seasonal cycle.

Following your suggestions, we have added these analyses and described them in the revised manuscript.

Line 448-452:

“To test if the phenological cycle affects our results, we repeated our analysis (1) using only peak growing season, the 3-month period with the maximum mean GPP across the available

years, where seasonal variability is muted; (2) using anomalies by removing the seasonal cycle, which was calculated by averaging all available years of the data and smoothing the series with a 30-day moving average as Feldman, Short Gianotti⁴. Both analyses yield similar results (Supplementary Fig. 9).”

The other observations I have are listed here below:

3.3 SWC is considered only until 30 cm and Water Table Depth has not been considered at all, but for trees these can be important and water access can happen even if the first layers of soil are dry. This should be at least discussed, better analysed if there are sites with these data.

Thank you for this suggestion. We found that grasslands and savannas show a more negative sensitivity of GPP to decreasing SWC than broadleaved deciduous forests and evergreen needle-leaved forests (Supplementary Fig. 5), which may be because forests can access to moisture in deeper soils and therefore have stronger resistance to drought^{5, 6, 7}.

There were 31, 24 and 17 sites with SWC measurements in the second (SWC_2), third (SWC_3), and fourth (SWC_4) depths, respectively (2-4: increases with the depth, 4 is deepest). We performed the same sensitivity analysis using these SWC observations from different layers, respectively, to evaluate the effects of SWC in different depths on the sensitivity of GPP to SWC and VPD (Fig. 2). The patterns of GPP sensitivity to SWC and VPD using deep SWC (SWC_2, SWC_3, SWC_4) are similar to the results obtained in our main analysis using the first layer of SWC observations, but we also found that there are greater negative sensitivities of GPP to both SWC decreases at dry soils and VPD increases at wet soils in SWC_4 than in other layers (Fig. 2). This suggests that it could cause more GPP reduction if the drought happens in deeper soil layers.

Fig. 2 The sensitivity of GPP to SWC (a-c) and VPD (d-f) using the SWC in the second (SWC_2, a, d), third (SWC_3, b, e), and fourth (SWC_4, c, f) depths, respectively (2-4: increases with the depth, 4 is deepest).

In the revised manuscript, we have added the discussion on deep SWC as suggested.

Line 181-184:

“Grasslands and savannas show a more negative sensitivity of GPP to decreasing SWC than broadleaved deciduous forests (DBF) and evergreen needle-leaved forests (ENF, Supplementary Fig. 5), which may be because forests can access to moisture in deeper soils and therefore have stronger resistance to drought^{5,6,7}.”

Line 445-448:

“To evaluate the effects of SWC in different depths on the sensitivity of GPP to SWC and VPD, we repeated the sensitivity analysis using 31, 24 and 17 sites with SWC measurements in the second (SWC_2), third (SWC_3), and fourth (SWC_4) depths, respectively (2-4: increases with the depth, 4 is deepest).”

Line 112-117:

“We also repeated our analysis using the SWC measurements from deep soil layers instead of the first layer (Methods). The patterns of GPP sensitivity to SWC and VPD using deep SWC are similar with the first layer, but we also found that there were greater negative sensitivities of GPP to both SWC decreases at dry soils and VPD increases at wet soils using the SWC in the deepest layer than in other layers (Supplementary Fig. 6). This suggests that it could cause more GPP reduction if the drought happens in deeper soil layers.”

3.4 Did you filtered and evaluated also the SWC and VPD quality? There are sites where SWC is in the first percentile (very dry) and VPD is very low and also sites with very high SWC and very high VPD. In some conditions this could be possible (the second case more difficult) but it is important to verify that it is not an artefact due to data quality issues.

Yes, we filtered and evaluated the SWC and VPD quality, too. SWC and VPD were quality controlled so that only measured and good-quality gap filled data (QC = 0 or 1) were used (Line 298-299). We also double checked the SWC and VPD data at each site.

Line 298-299:

“Data were quality controlled so that only measured and good-quality gap filled data (QC = 0 or 1) were used.”

3.5 The evaluation of the uncertainty in GPP was a good addition and it is convincing. It must be also noted (and may be reported) that for the estimation of GPP_DT the VPD is used as limiting factor so there is a risk of circularity. For this reason it was a good choice to use the GPP_NT. I think that the Standard Error is however not the best choice for the uncertainty analysis because the uncertainty is not normally distributes. I would suggest to evaluate the uncertainty in the NT using the quartile ranges, always available for all the sites in both the collections used and derived by the uncertainty in NEE.

Thank you for your suggestions. We have reported that it was a good choice to use the GPP_NT as suggested (Line 429-431).

Line 429-431:

“It should be noted that VPD is used as limiting factor for estimating GPP_DT, so it was a good choice to use the GPP_NT.”

Following your suggestions, we also repeated the sensitivity analysis using the quartile ranges of GPP (GPP_NT_VUT_25 and GPP_NT_VUT_75, Figs. 3a-d). The uncertainty in GPP sensitivities were quantified by calculating the differences of their sensitivities (e.g., GPP_NT_VUT_75 sensitivity to SWC minus GPP_NT_VUT_25 sensitivity to SWC) for each bin (Figs. 3e-f). We also calculated the relative uncertainty using the absolute value of differences of sensitivities divided by the absolute value of mean sensitivities, indicating that the uncertainty represents how many percent of the mean sensitivity (Figs. 3g-h).

Across all bins, we found that the differences of GPP sensitivity values obtained between the two GPPs were falling mostly in the range -0.1 to 0.1 (Figs. 3e-f), and relative uncertainties in most bins were less than 20% (Figs. 3g-h), indicating the uncertainty of NEE processing had minor effects on our results. Please note that the high levels of relative uncertainty occurred in the bins with statistically insignificant sensitivity values (Figs. 3a, c, g). Since these sensitivity values are close to zero, a low absolute uncertainty leads to a high relative uncertainty.

Fig. 3 Effect of the NEE processing on GPP sensitivity to SWC and VPD. a-b, The sensitivity of GPP to SWC (a) and VPD (b) using GPP_NT_VUT_25. c-d, The sensitivity of GPP to SWC (c) and VPD (d) using GPP_NT_VUT_75. e-f, Uncertainty in GPP sensitivity to SWC (e) and VPD (f). g-h, Relative uncertainty in GPP sensitivity to SWC (g) and VPD (h). Please note that the high levels of relative uncertainty occurred in the bins with statistically insignificant sensitivity values (a, c, g). Since these sensitivity values are close to zero, a low absolute uncertainty leads to a high relative uncertainty.

According to the reviewer’s suggestions, we have added the evaluation of the uncertainty in GPP using the quartile ranges in the revised manuscript.

Line 438-442:

“For the uncertainty of NEE processing, we repeated our analysis using the quartile ranges of GPP from the nighttime partitioning method (GPP_NT_VUT_25 and GPP_NT_VUT_75), which were available for all the sites in both the collections used and derived by the uncertainty in NEE. Similarly, the absolute and relative uncertainties from these two quartile GPPs were also quantified (Supplementary Fig. 8).”

Line 125-128:

“Concerning NEE processing, we repeated our analysis using the quartile ranges of GPP from the nighttime partitioning method (GPP_NT_VUT_25 and GPP_NT_VUT_75, Methods), and found that the differences of GPP sensitivity values obtained between the two quartile GPPs were pretty small in most bins (Supplementary Fig. 8).”

3.6 The reference to “extreme dry soil” (line 208) or SWC “extremely low” (line 214) is a little bit too strong in my opinion. The two effects (SWC and VPD) are basically equivalent from the 40th percentile of SWC, that means almost half of the range of values... I would not consider this “extreme”. In general I found the overall message a little bit too strong in highlighting the role of VPD respect to SWC while the data presented show that when limiting the SWC has also a strong role. Don’t you agree?

Yes, we agree and have modified the text as suggested (Line 210-217).

Line 210-217:

“Regarding the relative roles of SWC and VPD, we demonstrated that VPD dominates dryness stress on ecosystem production while SWC becomes important under dry soils. The mean linear regression slope (the standardized partial regression coefficient as all predictors were standardized) across all sites in Europe showed that VPD (−0.45 and −0.49, across 2014–2018 and 2014–2017, respectively) had larger negative effects on GPP than SWC (−0.22 and −0.14, Figs. 1d-e). Consistent with the linear model analysis in Europe, ANNs analysis found that VPD always dominates dryness stress on GPP as long as the SWC is not low, while the negative effects of decreasing SWC on GPP are larger than that of VPD under the low SWC conditions (< 30th percentiles, Fig. 3a).”

3.7 At line 289 it would be nice and useful to better clarify and report that all the corrections listed are already applied to the available product (so it has not been done by the authors for the specific study). This would promote the use of these open access data.

Following your suggestion, we have clarified them in the revised manuscript.

Line 289-294:

“Data were already processed following a consistent and uniform processing pipeline⁸. This data processing pipeline mainly included: (1) thorough data quality control checks; (2) calculation of a range of friction velocity thresholds; (3) gap-filling of meteorological and flux measurements; (4) partitioning of CO₂ fluxes into respiration and photosynthesis components; and (5) calculation of a correction factor for energy fluxes⁸. All the corrections listed were already applied to the available product⁸.”

3.8 Line 338: it would help to refer to the following section (page 23) where the methods is

introduced and explained.

We now referred to this section as the reviewer suggested.

3.9 At lines 342-343 you talk about “binned SWC and VPD anomalies into 10x10 percentile bins”. In fact you didn’t bin the anomalies but the SWC and VPD values (normalized, but I don’t think you can call them anomalies, it is the natural variability). I also suggest that the caption of Figure 2 should explain what are the percentiles in order to make it more self-explaining.

Thank you for your suggestions. We have modified this sentence to read “...then we binned daily SWC and VPD values into 10×10 percentile bins and assessed the sensitivities for each bin using ANNs for each site.”

In addition, we also explained the percentiles in the caption of Figure 2 as suggested. “The percentiles are the values of 10th, 20th, ..., and 90th percentile of SWC or VPD.”

3.10 On the Code availability it is stated that it is available upon request. I strongly suggest that at least the codes to calculate the G_c , A_{max} , and V_{cmax} should be openly shared. There could be (I’m sure) eddy covariance people that would like to use these codes and it is a good feedback toward people that shared the data for the paper. So please share the code in a repository and allow people to calculate the same parameters you estimated in the same way in order to do further analysis on the topic...

Thank you very much for this suggestion. We have shared these codes in a repository as suggested.

Line 503-504:

“The code used to calculate the G_c , A_{max} and V_{cmax} is publicly available at <https://github.com/fueco/GcAmaxVcmax>.”

References mentioned in the responses

1. Anderegg WR, *et al.* Hydraulic diversity of forests regulates ecosystem resilience during drought. *Nature* **561**, 538-541 (2018).
2. Liu L, Gudmundsson L, Hauser M, Qin D, Li S, Seneviratne SI. Soil moisture dominates dryness stress on ecosystem production globally. *Nat Commun* **11**, 4892 (2020).
3. Sulman BN, Roman DT, Yi K, Wang L, Phillips RP, Novick KA. High atmospheric demand for water can limit forest carbon uptake and transpiration as severely as dry soil. *Geophysical Research Letters* **43**, 9686-9695 (2016).
4. Feldman AF, Short Gianotti DJ, Trigo IF, Salvucci GD, Entekhabi D. Satellite-based assessment of land surface energy partitioning–soil moisture relationships and effects of confounding variables. *Water Resources Research* **55**, 10657-10677 (2019).
5. Konings AG, Gentile P. Global variations in ecosystem-scale isohydricity. *Global Change Biology* **23**, 891-905 (2017).
6. Martínez-Vilalta J, Garcia-Forner N. Water potential regulation, stomatal behaviour and hydraulic transport under drought: deconstructing the iso/anisohydric concept. *Plant, Cell & Environment* **40**, 962-976 (2017).
7. Teuling AJ, *et al.* Contrasting response of European forest and grassland energy exchange to heatwaves. *Nature Geoscience* **3**, 722 (2010).
8. Pastorello G, *et al.* The FLUXNET2015 dataset and the ONEFlux processing pipeline for eddy covariance data. *Scientific data* **7**, 1-27 (2020).

REVIEWERS' COMMENTS

Reviewer #3 (Remarks to the Author):

Thanks for considering the suggestions. Good work, congratulation, for me the paper can be accepted as it is.

**Atmospheric dryness reduces photosynthesis along a large range of soil water deficits
NCOMMS-21-22710B**

Response to Reviewers

Reviewer #3 (Remarks to the Author):

3.1 Thanks for considering the suggestions. Good work, congratulation, for me the paper can be accepted as it is.

We thank the reviewer for supporting our study and appreciate the constructive comments in the process.